# Potent DNA gyrase inhibitors bind asymmetrically to their target using symmetrical bifurcated halogen bonds

Anja Kolarič[1,2], Thomas Germe [3], Martina Hrast [2], Clare E. M. Stevenson [3], David M. Lawson [3], Nicolas P. Burton [4], Judit Vörös [3], Anthony Maxwell [3], Nikola Minovski [1✉] & Marko Anderluh [2✉]

Novel bacterial type II topoisomerase inhibitors (NBTIs) stabilize single-strand DNA cleavage breaks by DNA gyrase but their exact mechanism of action has remained hypothetical until now. We have designed a small library of NBTIs with an improved DNA gyrase-binding moiety resulting in low nanomolar inhibition and very potent antibacterial activity. They stabilize single-stranded cleavage complexes and, importantly, we have obtained the crystal structure where an NBTI binds gyrase–DNA in a single conformation lacking apparent static disorder. This directly proves the previously postulated NBTI mechanism of action and shows that they stabilize single-strand cleavage through asymmetric intercalation with a shift of the scissile phosphate. This crystal stucture shows that the chlorine forms a halogen bond with the backbone carbonyls of the two symmetry-related Ala68 residues. To the best of our knowledge, such a so-called symmetrical bifurcated halogen bond has not been identified in a biological system until now.

[1] Theory Department, Laboratory for Cheminformatics, National Institute of Chemistry, Hajdrihova 19, 1001 Ljubljana, Slovenia. [2] Department of Pharmaceutical Chemistry, Faculty of Pharmacy, University of Ljubljana, Aškerčeva 7, 1000 Ljubljana, Slovenia. [3] Department of Biological Chemistry, John Innes Centre, Norwich Research Park, Norwich NR4 7UH, UK. [4] Inspiralis Ltd., Innovation Centre, Norwich Research Park, Colney Lane, Norwich NR4 7GJ, UK. ✉email: nikola.minovski@ki.si; marko.anderluh@ffa.uni-lj.si

The fact that bacterial DNA gyrase remains one of the most studied antibacterial targets is easy to justify. A simple literature query on the PubMed repository returns approximately 200 hits annually for the past decade with January 2020 giving 30 hits already[1]. A Protein Data Bank (PDB)[2] query with the same search terms gives 190 different crystal structures and a very recent cryo-electron microscopic structure of the complete *Escherichia coli* DNA gyrase in a ternary complex with DNA and a small molecule GyrA inhibitor gepotidacin[3]. The reason for this popularity is clear: DNA gyrase is an essential bacterial type II topoisomerase that is involved in the maintenance of the correct spatial DNA topology in bacteria[4]. Moreover, it has been a validated antibacterial target for decades, being the target of fluoroquinolone antibacterials[5]. DNA gyrase consists of two copies of GyrA (which contains the catalytic tyrosine) and two copies of GyrB (which comprises the ATPase activity) thus functioning as an $A_2B_2$ heterotetramer[4]. The $A_2B_2$ heterotetramer can accommodate a variety of inhibitors that prevent DNA gyrase function, namely, the catalytic inhibitors and cleavage-complex stabilizers[5–8]. Among the recently discovered compounds, the novel bacterial type II topoisomerase inhibitors (NBTIs) are probably the closest to clinical use[9].

The NBTIs form a gyrase–DNA–inhibitor ternary complex (as demonstrated by *Staphylococcus aureus* DNA gyrase)[10] and have a somewhat similar intercalating mechanism of action to fluoroquinolones with a single inhibitor molecule bound centrally between the two scissile DNA bonds and in a pocket between the two GyrA subunits, as demonstrated for gepotidacin[10,11]. According to their mechanism of action, NBTIs are composed of the intercalating "left-hand side" (LHS) and the GyrA binding "right-hand side" (RHS) linked with an appropriate spacer (Fig. 1a)[5]. In contrast to fluoroquinolones, gepotidacin stabilizes only single-strand cleavage breaks, which is consistent with the LHS having an asymmetrical binding mode, i.e., it can bind in two conformations that are related by a 180° rotation within the same crystal. This is evident in the crystal structure of GSK299423 (**1**) with *S. aureus* gyrase and a DNA fragment (PDB ID: 2XCS) where the compound sits on the twofold axis and is not C2 symmetric[7,10–12]. Although this mechanism seemed very probable, until now the exact conformation of DNA in this asymmetric complex was difficult to ascertain due to the lack of

crystal structure with compound and DNA in a single orientation. Namely, all of the previously published crystal structures "suffer" from static disorder (Fig. 1b), so it has not yet been possible to correlate a compound orientation to a DNA orientation[7].

In this work, we present a small library of NBTIs with an improved RHS part of the molecule. Although the RHS binding GyrA interface pocket lacks polar amino acid residues and thus specific interactions, we show that compounds having a simple RHS may form either hydrogen or halogen bonds with the GyrA interface backbone (Fig. 1c). The most potent compounds of the series inhibit gyrase in a low nanomolar concentration range and have very potent antibacterial activity, as displayed by their $IC_{50}s$ (the concentration of inhibitor where the residual activity of the enzyme is 50%) and minimal inhibitory concentrations (MICs), respectively (Table 1). Further exploration of their mechanism of action by cleavage assays confirms that compounds stabilize single-strand cleavage breaks. The crystal structure of one of our inhibitors in a ternary complex with a gyrase-nicked DNA fragment reveals that the crystallized inhibitor interacts with the GyrA heterodimer by a symmetrical bifurcated halogen bond. Finally, we note that, in this crystal structure the NBTI binds in a single conformation devoid of apparent static disorder, thus directly proving the previously postulated, but not completely verified, mechanism of action.

## Results

**Design of NBTIs with the focus on the RHS**. As presented in Fig. 1a, the structure–activity relationship of NBTI's LHS and linker moieties is mostly well established[5], while the RHS was previously underexplored in terms of achieving additional interactions. Namely, the RHS binding pocket is delineated by rather hydrophobic amino acid residues (i.e., Ala68, Gly72, Met75, Met121, lying on the GyrA α3 helices). These residues were previously recognized as key target points for establishing GyrA-NBTI van der Waals interactions only. In contrast with previous designs, we saw an opportunity to go a step further in designing a series of NBTIs capable of targeting the backbone carbonyl oxygen of both neighbouring Ala68 residues, thereby strengthening their binding and consequently improving their antibacterial potency. Furthermore, the backbone carbonyls cannot be removed by simple mutations and the NBTIs forming

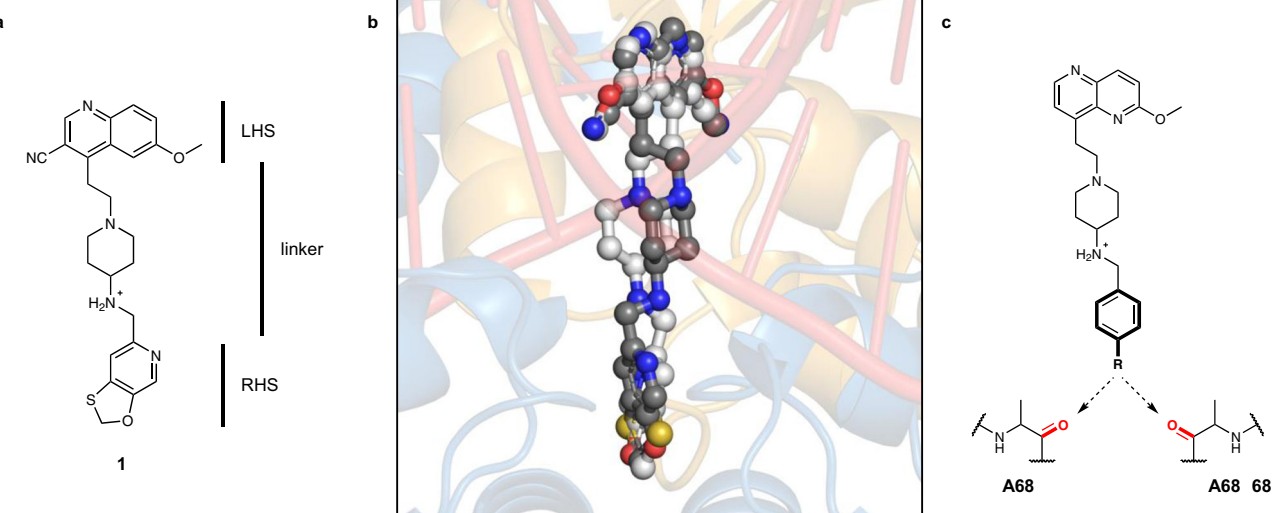

**Fig. 1 NBTI structure and apparent static disorder in DNA gyrase-binding site. a** Structure of the representative NBTI GSK299423 (**1**), indicating the main structural features of the molecule: the "left-hand side" (LHS) and the "right-hand side" (RHS) connected by linker[5,10]. **b** Two conformations of GSK299423 (represented as balls and sticks colour coded by element) (**1**) modelled as a superimposition of the two orientations due to static disorder[7,10]. **c** Our design strategy includes a simple phenyl RHS (represented by bold bonds), substituted in such a manner to allow binding by either halogen or hydrogen bonding with the backbone carbonyl oxygen of one of Ala68.

**Table 1 DNA supercoiling inhibition of *S. aureus* and *E. coli* DNA gyrase and decatenation activity of human topoisomerase IIα affected by NBTIs.**

| Cmpd | Structure | DNA gyrase IC$_{50}$ [μM][a] | | MIC [μg/mL] | | | | | Human Topo IIα [% RA][g] |
|---|---|---|---|---|---|---|---|---|---|
| | | *S. aureus* | *E. coli* | *S. aureus*[b] | MRSA[c] | *E. coli*[d] | *E. coli* D22[e] | *E. coli* N43[f] | |
| 2 | | 1.02 ± 0.20 | 40.60 ± 1.97 | 4 | 8 | 64 | 16 | 8 | 97 ± 10 |
| 3 | | 0.55 ± 0.06 | 16.95 ± 1.69 | 1 | 2 | 32 | 16 | 4 | 101 ± 11 |
| 4 | | 0.035 ± 0.14 | 1.71 ± 0.05 | 0.125 | 0.25 | 4 | 2 | 0.5 | 102 ± 1 |
| 5 | | 0.007 ± 0.000[h] | 0.57 ± 0.06 | 0.031 | 0.062 | 2 | 0.125 | 0.125 | 98 ± 3 |
| 6 | | 0.011 ± 0.003[h] | 0.28 ± 0.02 | 0.0078 | 0.015 | 2 | 0.125 | 0.078 | 88 ± 3 |
| 7 | | 4.39 ± 0.55 | 56.56 ± 3.32 | 4 | 4 | 128 | 16 | 16 | 98 ± 1 |
| 8 | | 1.82 ± 0.11 | >100 | 2 | 4 | >128 | 128 | 16 | 98 ± 2 |
| 9 | | 0.067 ± 0.01 | 11.89 ± 0.83 | 0.125 | 0.25 | 32 | 8 | 1 | 94 ± 1 |

[a]IC$_{50}$ value represents the mean value of two to four independent measurements.
[b]*S. aureus* ATCC 29213.
[c]*S. aureus* NCTC12493.
[d]*E. coli* ATCC 25922.
[e]*E. coli* D22 bears a mutation in the lpxC gene that increases membrane permeability.
[f]*E. coli* N43 AcrA knockout strain (knockout of cell membrane efflux pump).
[g]The mean ± SD percentage of residual activity of the enzyme at compound concentration 10 μM obtained by two independent measurements.
[h]IC$_{50}$s were determined by gel-based assays. A slightly different method was used for the final readout IC$_{50}$ determination, as explained in "Methods".

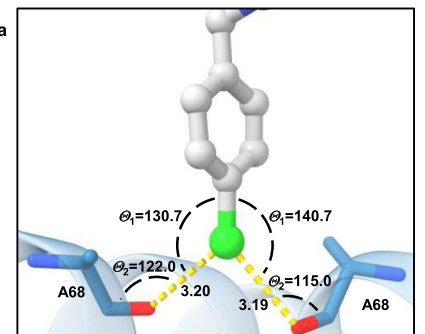 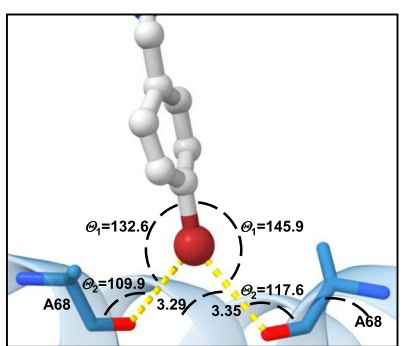 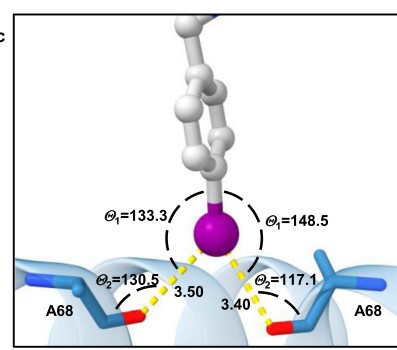

**Fig. 2 Prediction of halogen bonding for compounds 4–6.** Simulation snapshots depicting the ability of halogen bonding propensity for compounds **4**–**6** by averaging MD resulted interatomic X···O distances [Å] and $\theta_1$ [C-X···O], i.e., $\theta_2$ [X···O=C] angles calculated over the entire simulation trajectories between backbone carbonyl oxygen of Ala68 (represented as sticks colour coded by element) from both GyrA subunits (cartoon representation in blue; PDB ID: 2XCS)[10] and our studied *p*-halogenated RHSs: **a** *p*-chlorophenyl (**4**, chlorine atom in green); **b** *p*-bromophenyl (**5**, bromine atom in red); and **c** *p*-iodophenyl (**6**, iodine atom in violet).

interactions with these carbonyls should be less prone to the development of bacterial resistance. This structural feature of Ala68 was already previously recognized as a promising partner for establishing atypical hydrogen bonds with RHS moieties. Namely, in the crystal structure of GSK299423 (**1**) (PDB ID: 2XCS), such an unusual hydrogen bond was observed between the methylene group of oxathiolopyridine ring and Ala68 backbone carbonyl oxygen in DNA gyrase[10]. We have previously designed NBTIs with augmented RHSs that would allow potent interaction within the RHS binding pocket[13]. However, they were only partially successful and we reasoned that the lower than expected potency was due to bicyclic or conjugated biaryl nature of the designed RHSs that were simply too bulky to be accommodated by the RHS binding pocket. Herein, our idea was to design such RHS moieties that would contain a sterically less demanding phenyl moiety substituted in such a manner to strengthen binding by either another type of hydrogen bonding or halogen bonding with the backbone carbonyl oxygen of at least one of the two Ala68 residues. Halogen bonding, a non-covalent interaction of halogen atoms, is explained by the presence of a region of positive electrostatic potential, the so-called σ-hole, on the outermost portion of the halogen's surface, centred on the R–X axis (X = halogen, R = alkyl or aryl carbon). According to Clark et al., in molecules that contain Cl, Br, and I atoms, the halogen atoms closely approximate the s2p2 *xp2 yp1 z* configuration, where the *z*-axis is along the R–X bond[14,15]. Due to spin around the R–X bond, the three unshared electron pairs form a doughnut-like cloud of negative electrostatic potential around the halogen atom, leaving the positive potential region at its apex, the σ-hole. The σ-hole differs in size according to the halogen involved (I > Br > Cl >> F) and offers a possibility for an interaction with a Lewis base, e.g. a lone electron pair of a heteroatom like a carbonyl oxygen. The size of the σ-hole defines the allowed X···O bond lengths (Cl···O < 3.27 Å, Br···O < 3.37 Å, I···O < 3.50 Å) and angles ($140° \leq \theta_1$ [C-X···O] $\leq 180°$ and $90° \leq \theta_2$ [X···O=C] ~ 120°)[16]. With these ideas outlined, phenyl RHS moieties substituted with halogens (-F, -Cl, -Br, -I), H-bond donors (-CH₂OH, -CONH₂), or a moiety capable of dipolar interactions (–NMe₂) in the *para*-position were preferred. Virtual NBTIs were constructed and, since LHS and linker were not in the focus of our structural alteration, we opted to retain the well-established methoxy-naphthyridine LHS and aminopiperidine central unit (Fig. 1c and Table 1)[10]. Prior to the synthesis of our NBTIs with the aforementioned RHS moieties, the binding poses of such assembled NBTI constructs were initially predicted by flexible molecular docking calculations within GyrA-NBTIs' binding site of *S. aureus* (PDB ID: 2XCS). Taking into consideration the allowed X···O bond lengths and angles, compounds with *p*-chlorophenyl (**4**), *p*-bromophenyl (**5**), and *p*-iodophenyl (**6**) RHSs displayed a high probability of halogen bonding interactions

with the backbone carbonyl oxygen of the Ala68 residue (C-X···O=C) from at least one GyrA subunit (Supplementary Fig. 1 and Supplementary Table 2). This was not the case with hydrogen bond and dipolar interaction-forming moieties.

To further investigate the behaviour of our NBTI series as well as the propensity of these predicted halogen bonding interactions, the compounds underwent molecular dynamics (MD) simulations. The X···O distances [Å] and $\theta_1$ [C-X···O], i.e., $\theta_2$ [X···O=C] angle distribution plots related to the halogen bonding propensity between *p*-substituted halogens of investigated NBTIs (**3**–**6**) and the backbone carbonyl oxygen of the Ala68 residues from both GyrA subunits was thoroughly examined over the entire MD simulation trajectory (Supplementary Fig. 3). The X···O distance distribution plots suggest that in all cases *p*-substituted halogen has a high affinity to establish a halogen bonding interaction with the Ala68 backbone carbonyl oxygen of at least one GyrA subunit. While the frequency of Cl···O (**4**) distances to both Ala68 carbonyl oxygens is almost balanced (Supplementary Fig. 3b), one can observe a significant increment in the frequency of Br···O (**5**) and I···O (**6**) distances, however, mainly limited to targeting the Ala68 backbone carbonyl of a sole GyrA subunit (Supplementary Fig. 3c, d). Nevertheless, the ability of halogen bonding interactions for our series of compounds (**4**–**6**) is noticeable if one takes into consideration the portion of distance bars that are covered by the grey shaded areas. These areas represent the allowed interatomic X···O distance ranges (Cl···O < 3.27 Å, Br···O < 3.37 Å, I···O < 3.50 Å) that increase with the halogen atom ability to form a halogen bond (Fig. 2, Supplementary Fig. 3, and Supplementary Table 3)[16,17]. A similar conclusion can be reached by observing the distribution plots related to $\theta_1$ [C-X···O] and $\theta_2$ [X···O=C] angles, which are nearly equalized (Supplementary Fig. 3, middle and last column of the distribution plots). Additionally, this supports the hypothesis that our series of compounds **4**–**6** might be capable to establish almost symmetrical bifurcated halogen bonds with electron-rich backbone carbonyl oxygens of both Ala68 residues as well. The latter is particularly interesting in the case of $\theta_2$ [X···O=C] angles (Supplementary Fig. 3b–d), whose distribution plots are practically overlapped indicating to a possibility that the σ-hole on *p*-substituted halogens in our compounds might be effectively reached (Fig. 2, Supplementary Fig. 3, and Supplementary Table 3)[15].

**Inhibition of gyrase-catalyzed DNA supercoiling and antibacterial activity.** As shown in Table 1, all designed and synthesized compounds (for synthesis details, please see Supplementary Information) exhibited stronger inhibition of *S. aureus* DNA gyrase

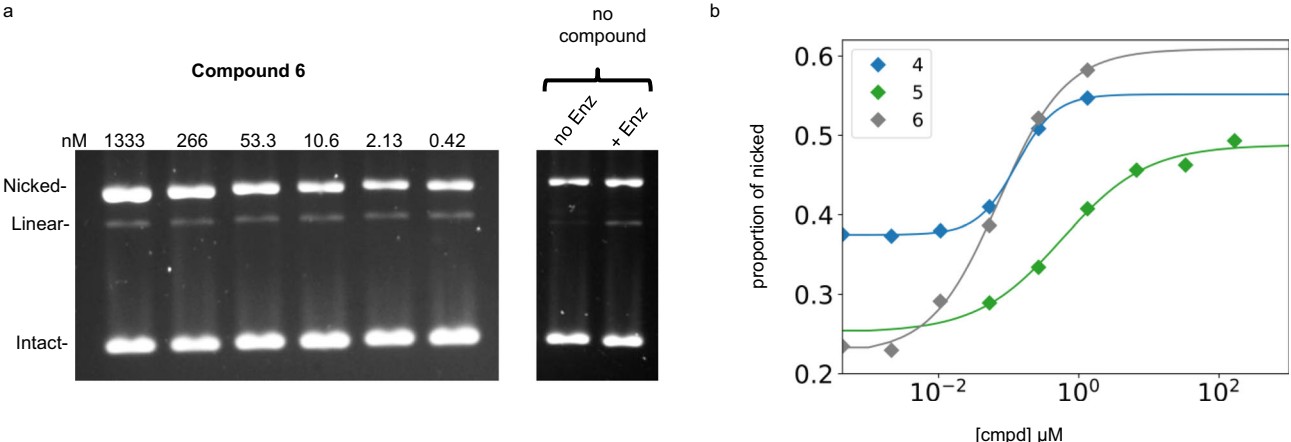

**Fig. 3 Cleavage assay with *E. coli* DNA gyrase.** Cleavage complexes formed by **4–6** were trapped with SDS (see "Methods") before treatment with Proteinase K. The cleaved plasmids were resolved by agarose gel electrophoresis. **a** Gel image for **6**; increasing amount of compound results in increasing nicked DNA being trapped. No increase in linear plasmid was observed. **b** The gel for compound **6** and similar gels for **4** and **5** (Supplementary Fig. 8) were quantified by densitometry. The proportion of nicked to the total amount of DNA in the lane was plotted against the compound concentration in **b**. For each compound, the data were fitted to the Hill equation. The $EC_{50}$ parameter for the best fit is the measured $CC_{50}$. The cleavage assays were reproduced three times, while the no enzyme control was reproduced twice with similar results. Uncropped versions of gel images are available as Supplementary Figs. 22 and 23.

compared to *E. coli* and, accordingly, more potent antibacterial activities (Table 1), which is in line with our design that focused on DNA gyrase from *S. aureus*. As we anticipated, *p*-halogenated phenyl RHSs (**3–6**) turned out to be a good selection, since *p*-fluoro phenyl RHS (**3**) already enhances gyrase inhibition and improves antibacterial potency for both *S. aureus* and *E. coli*, compared to an NBTI with an unsubstituted phenyl RHS (**2**). By introducing a *p*-chloro phenyl (**4**), $IC_{50}$s in the low nanomolar range were achieved for *S. aureus* and eightfold lower MICs were observed on all three strains. The breakthrough was accomplished with *p*-bromo (**5**) and *p*-iodo phenyl (**6**) derivatives that show strong inhibitory activities against DNA gyrase from both organisms (*S. aureus* and *E. coli*). *S. aureus* DNA gyrase inhibitory potencies ($IC_{50}$ of 0.007 μM for **5** and 0.011 μM for **6**, respectively) are the most potent reported to date for an NBTI. These results demonstrate that our design strategy was successful and that the potency increases with the *p*-substituted halogen size F << Cl < Br ~ I. The latter is highly indicative of the presence of halogen bonds formed by Cl, Br, and I substituents[17]. The same trend of enzyme inhibition translates into notable antibacterial activities against *S. aureus* and Methicillin-resistant *S. aureus* (MRSA; Table 1). The only exception is the Br–I pair, where potency is virtually the same (see "Discussion").

We also investigated substitution of phenyl with some other substituents, aiming for the formation of hydrogen bonds or dipolar interactions with the carbonyl of Ala68. A compound with *p*-hydroxymethyl (**7**) led to a significant decrease of potency and basically the same result was observed for a *p*-carboxamide phenyl (**8**) derivative, with both compounds losing potency on *E. coli* gyrase as well. Interestingly, compound **9**, comprised of *p*-dimethylaminophenyl RHS, was highly active against *S. aureus* gyrase ($IC_{50}$ = 0.067 μM) with reasonable antibacterial potency against MRSA (MIC = 0.25 μg/mL) and *S. aureus* (MIC = 0.125 μg/mL). Target inhibition constants and MIC values on *E. coli* were in general higher showing a decline in the observed potency compared to *S. aureus*. To elucidate the cause of this observation, we have measured MICs on *E. coli* strains with knocked out AcrA efflux pumps (N43) and the *E. coli* strain with the mutation in the lpxC gene that increases membrane permeability (D22, data in the Table 1). The comparison of MICs reveal that our compounds are effectively pumped by efflux pumps, while membrane permeability is not an issue.

All of the compounds confirmed selectivity for bacterial gyrase over orthologous human topoisomerase IIα, even the most active compound **5** showed at least 1000-fold higher selectivity for bacterial enzyme over the human one thus supporting our compounds potential for selective toxicity towards bacteria. We have subjected the compounds to preliminary cytotoxicity studies on human HUVEC and HepG2 cell lines and determined their hERG (human ether-a-go-go-related gene) potassium channel inhibition (Supplementary Table 7). The results point to safety issues related primarily with hERG inhibition, which is a class-related problem for NBTIs, although successful cases devoid of hERG issues have been reported and some are in clinical trials[5]. Having this in mind, our compounds should be regarded as functional molecular probes for investigating the mechanism of DNA gyrase inhibition, while hit-to-lead optimization will be done to expand the current NBTI library and yield candidates with less toxicity issues suitable for in vivo studies.

**Compounds 4–6 stabilize single-stranded cleavage complexes with *E. coli* DNA gyrase.** To determine the potency of the compounds in stabilizing cleavage complexes, we performed cleavage assays with *E. coli* DNA gyrase. In these assays, sodium dodecyl sulfate (SDS) is added to cleavage complexes formed with plasmid DNA to denature the protein, thereby rendering DNA cleavage irreversible. The proteins that are covalently attached to the cleaved plasmids are subsequently digested with proteinase K, allowing analysis of the free cleaved plasmids by agarose gel electrophoresis to determine the nature and extent of cleavage. This is a direct reflection of the amount of cleavage complex formed by the compound, DNA, and enzyme. We performed this assay with all three compounds and observed the trapping of only nicked DNA (Fig. 3), showing that all three compounds stabilize single-strand cleavage complexes. We also examined the amount of cleavage complexes trapped in response to compound concentration. This allowed us to measure the $CC_{50}$, the compound concentration needed to stabilize half the amount of cleavage complex at saturation. For all three compounds, the $CC_{50}$ was found to be in the sub-μM range, consistent with the $IC_{50}$ measurements and with the idea that the formation of cleavage complexes results in enzyme inhibition.

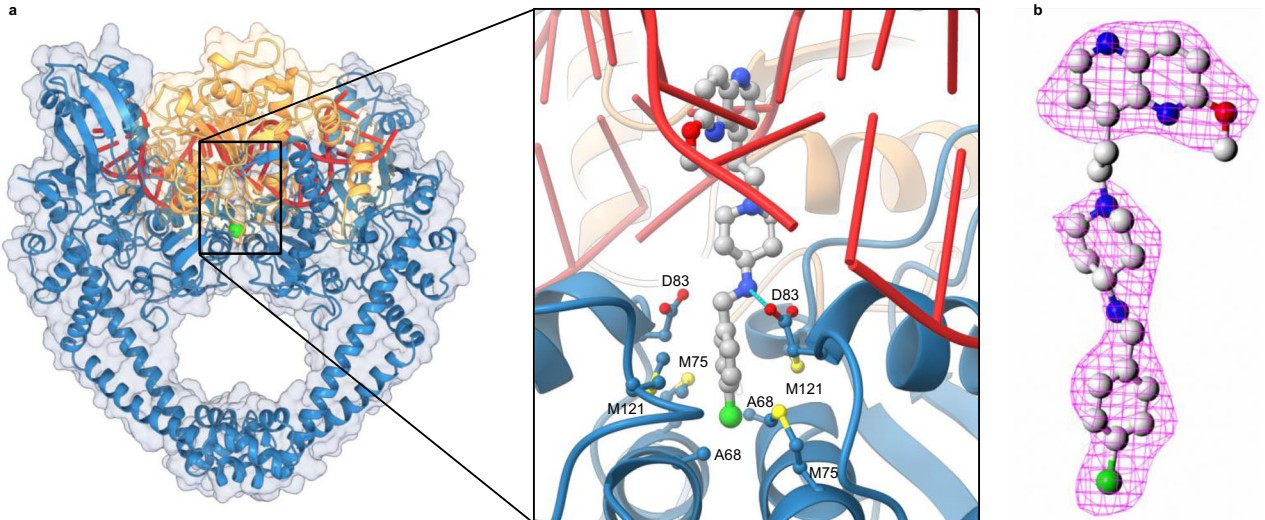

**Fig. 4 The *S. aureus* DNA gyrase in complex with 4 and DNA. a** The 2.3-Å resolution crystal structure of compound **4** bound to a DNA–gyrase complex (PDB ID: 6Z1A). GyrA and GyrB are represented as cartoons in blue and yellow, respectively. **4** is represented as balls and sticks colour coded by elements and DNA as cartoon in red. The single hydrogen bond to the ligand is shown as blue dots. **b** Omit $mF_{obs} - DF_{calc}$ difference electron density for the bound ligand (magenta mesh) calculated at 2.3-Å resolution and contoured at 1.6σ. This shows clearly the asymmetry of the compound, including the intercalating moiety.

The level of cleavage at saturation will depend on the enzyme-to-DNA ratio and the proportion of cleaved compound–enzyme–DNA complexes. When we assayed our preparation of *E. coli* gyrase with the fluoroquinolone ciprofloxacin, a much higher level of cleavage (double-strand cleavage) was observed at saturation. This is due to the fact that probably all the ciprofloxacin–enzyme–DNA complexes are cleaved, unlike with our compounds. Using ciprofloxacin-induced cleavage as a reference[18], we have found that up to approximately 25% of complexes are cleaved. This is consistent with other NBTI compounds that bind in the same pocket[7].

**Structure of the *S. aureus* gyrase–DNA–compound 4 ternary complex.** It has been suggested that the reason compounds binding in the NBTI pocket stabilize single-strand cleavage is connected to their asymmetric binding mode[7]. In previous NBTI structures, the compound–enzyme–DNA complexes usually adopt, within the asymmetric unit, each of the two orientations around the C2 axis randomly across the crystal. This results in static disorder of any C2 asymmetry within the complex, including NBTI compounds, which are therefore modelled as a superimposition of the two orientations. It is therefore impossible to correlate one orientation of the compound to the structure of the cleavage catalytic pocket since any asymmetry within the complex will be seen as an average[7,10,12].

To study the exact binding mode of the gyrase inhibitors, we have conducted crystallization studies with halogenated compounds **4–6**. Here we report the 2.3-Å resolution crystal structure of a potent compound (**4**) in a complex with the *S. aureus* DNA gyrase core fusion and DNA. Significantly, this structure did not suffer from the static disorder seen previously[7], revealing an unambiguous picture of the relationship between the compound, the protein, and the DNA. The LHS (methoxy-naphthyridine) of compound **4** intercalates between DNA base pairs on the twofold axis of the complex, midway between the two DNA cleavage sites, and *p*-chlorophenyl RHS moiety sits in a pocket between the two GyrA subunits.

Crystallization of the ternary complex in a single orientation across the crystal allows the correlation of compound orientation with the conformation of the catalytic pocket. The positive omit electron density (Fig. 4b) clearly shows the asymmetry of

compound **4**. In addition, the catalytic pockets display two different configurations. On one side, the single catalytic metal, in this case a manganese is coordinated by Asp508, Glu435, and the scissile phosphate, which is not bonded to the 3′-OH as the crystallized DNA is doubly nicked at the cleavage sites. This is close to the A-configuration and presumably reflects a catalytically active conformation due to the metal–phosphate contact. On the other side, the phosphate is more remote from the catalytic metal, which adopts the B-configuration, coordinated by Asp508, Asp510, and water molecules[7].

**The LHS binding pocket: the asymmetry of the compound induces asymmetry of DNA conformation and in turn asymmetry of catalytic pocket configuration.** To try to explain why two different conformations are adopted and how this relates to the orientation of the compound, we superposed the two sides, using as a reference the secondary structure involved in the formation of the catalytic pocket WHD (winged helix domain) and TOPRIM (topoisomerase primase domain) and contacting the DNA away from the cleavage site (Tower domain, Supplementary Fig. 4). It is important to note that the DNA itself is not used as a reference in this superimposition. One moiety of the compound intercalates between base +2 and base +3 (counting 5′ to 3′ from the scissile phosphate). This intercalation is presumably stabilized by stacking interactions. The intercalating moiety is asymmetric (Fig. 4b), which would affect the stacking interaction of the neighbouring bases differently depending which side they are on. Comparing bases with their C2 symmetry mate, a rotation is observed for base +1 to +4, i.e. two bases on each side of the compound. This presumably reflects asymmetric (with respect to the C2 axis) stacking interactions of the bases adjacent to the intercalating moiety. However, this rotation is absent for bases that are more distant from the compound (+5 to +8 and further, Fig. 5b). This rotation of the bases is accompanied by a shift in the position of the cognate riboses and therefore the phosphates attached also shift their position compared to their C2 symmetry mates (Fig. 5a, red arrows) and this includes the scissile phosphate. We therefore conclude that, because the intercalating moiety of the compound is asymmetric, intercalation results in

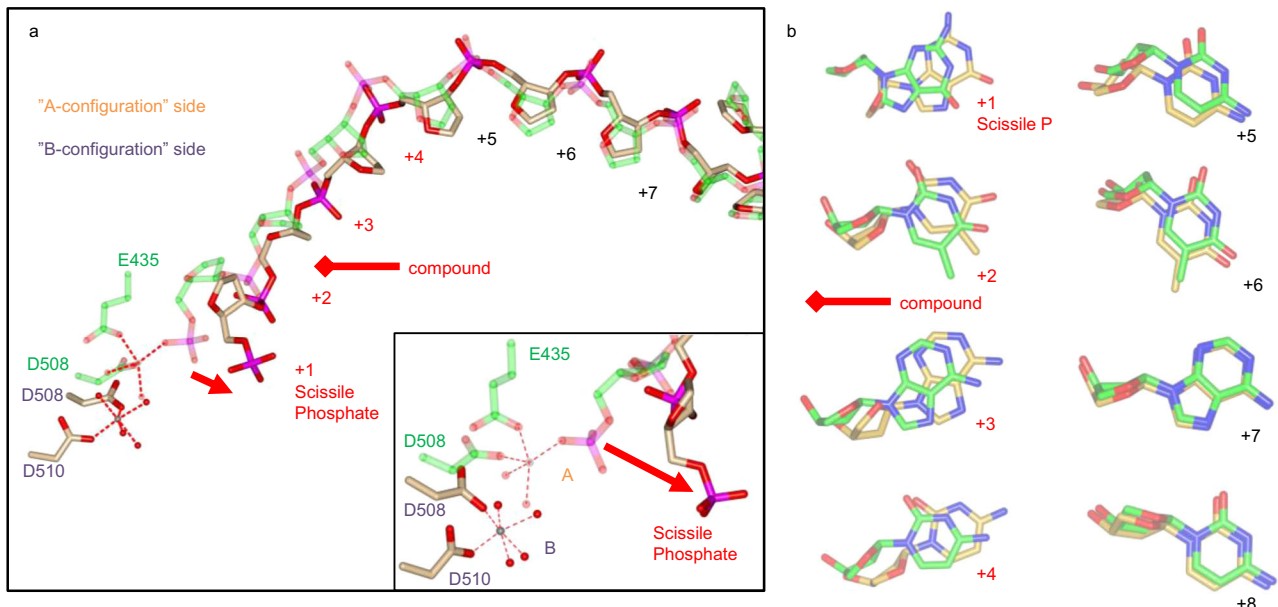

**Fig. 5 Asymmetry of compound binding correlates with an asymmetry of catalytic pocket configuration. a** Superimposition of the two 5′ DNA strands with the tower domain and the WHD domain as references. For clarity only, the phosphate and ribose backbone is shown. This shows the two configurations (A and B) adopted by the metal (see inset). On one side, the A metal contacts the scissile phosphate. On the other side, the scissile phosphate is away from the metal, which is in the B configuration. The superimposition also shows the shifting of the phosphate from nucleotide +1 (scissile phosphate) to +4, around the compound (red arrows). **b** The shifting of the phosphate position correlates with a rotation of the bases around the double-helix axis observed only for two nucleotides on each side of the compound. One DNA strand is coloured in green, the complementary strand in tan, while O, N, and P are colour coded by element.

rotation of the base adjacent on one side compared to the C2 symmetry-mate base on the other side. The next base over is also affected and therefore the phosphates positioned 2 nucleotides away from the compound (nucleotides +1 to +4) are affected. This moves the scissile phosphate away from the metal, which then adopts the B-configuration (Fig. 5a). Presumably this blocks resealing on this side by keeping the cleaved phosphate away from the catalytic metal, preventing the reverse phosphotransfer reaction. This is consistent with the view that the position of the scissile phosphate with regards to the catalytic metal determines whether phosphotransfer occurs (see "Discussion").

The methoxy group bound to naphthyridine intercalating LHS of **4** importantly influences the observed DNA asymmetry. It shifts the adjacent thymine away thus pushing the whole strand of the DNA (Fig. 6a, b: right down to the naphthyridine moiety). This influences not only the intercalator–base pairs stacking interactions (Fig. 6a) but also the base pairing of the affected thymine by hydrogen bonding as well (Fig. 6b) and allows the asymmetry in DNA that protrudes to the scissile phosphate position 2 nucleotides away from the compound.

**The _p_-halogenated phenyl fits into RHS binding pocket.** As depicted in the Fig. 4a, the _p_-chlorophenyl RHS moiety is tightly embraced by the surrounding amino acid residues in a tight pocket between the two GyrA subunits. This pocket is rather hydrophobic and consists of amino acids Ala68, Gly72, Met75, and Met121, which are forming mainly van der Waals interactions with RHS moiety. The major binding driving force for the RHS, however, is most probably the observed halogen bonding interaction between the chlorine and the Ala68 backbone carbonyl oxygen of both GyrA subunits (Figs. 4a and 7). This specific symmetrical bifurcated halogen bonding is further explicated in the "Discussion" section.

## Discussion

The design of our NBTIs relied on an assumption that a simple _p_-substituted RHS would allow either hydrogen or halogen bonding with the GyrA backbone interface (Fig. 1c). The most potent compounds of the series with halogens at the _p_-position inhibit gyrase in the low nanomolar concentration range and display powerful antibacterial activity, proving that our design was fruitful. By using cleavage assays with _E. coli_ gyrase, we have proven that our compounds stabilize single-stranded cleavage complexes, as expected for NBTIs.

A genuine breakthrough in understanding the mechanism of the NBTI mode of action was made possible by obtaining the crystal structure of one of our inhibitors (**4**) in a ternary complex with gyrase and a doubly nicked DNA fragment. To our great pleasure, this is the first crystal structure where an NBTI binds in a single conformation lacking static disorder and thus directly proves the previously postulated mechanism of action of NBTIs. Our conclusion is that the intercalation of the asymmetric LHS of compound **4** between the two central base pairs results in a rotation around the axis of the double helix of each base pair in the same direction, thus breaking the C2 symmetry. This is evidenced in the superimposition of the two 5′ strands with reference to the C2-symmetric Tower, WHD, and TOPRIM domains (Fig. 5 and Supplementary Fig. 4). The rotation results in a shift of the DNA backbone away from the cleavage site on one side only. This effect is visible up to two bases away from the compound. Consequently, the scissile phosphate is separated from the catalytic metal on one side only. In this structure, the DNA is doubly nicked and therefore the scissile phosphates are able to move away from the 3′ OH. When the DNA is uncleaved, this effect would presumably occur as well and only allows the scissile phosphate to get close to the catalytic metal on one side only, thus explaining why the compound stabilizes single-strand cleavage. It is expected that almost any intercalating moiety could establish

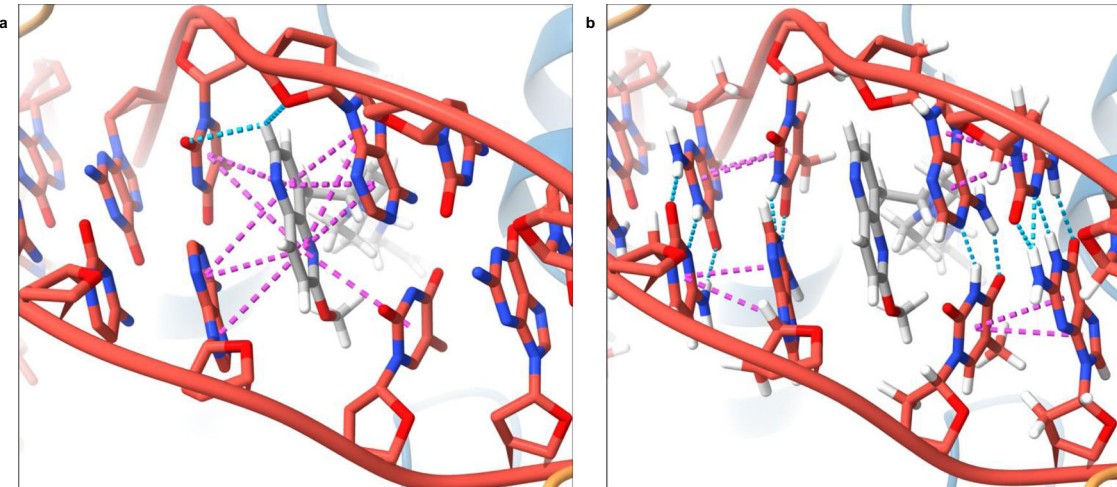

**Fig. 6 Intercalation of LHS between symmetrical DNA base pairs. a** Stabilization of asymmetric LHS of **4** (sticks representation colour coded by element) by stacking interactions causes uneven rotation of surrounding DNA base pairs (cartoon representation in red), thus breaking DNA symmetry. Stacking interactions are represented as magenta dots, while pseudo hydrogen bonds are represented as blue dots. **b** Interaction network of DNA base pairs affected by LHS intercalation.

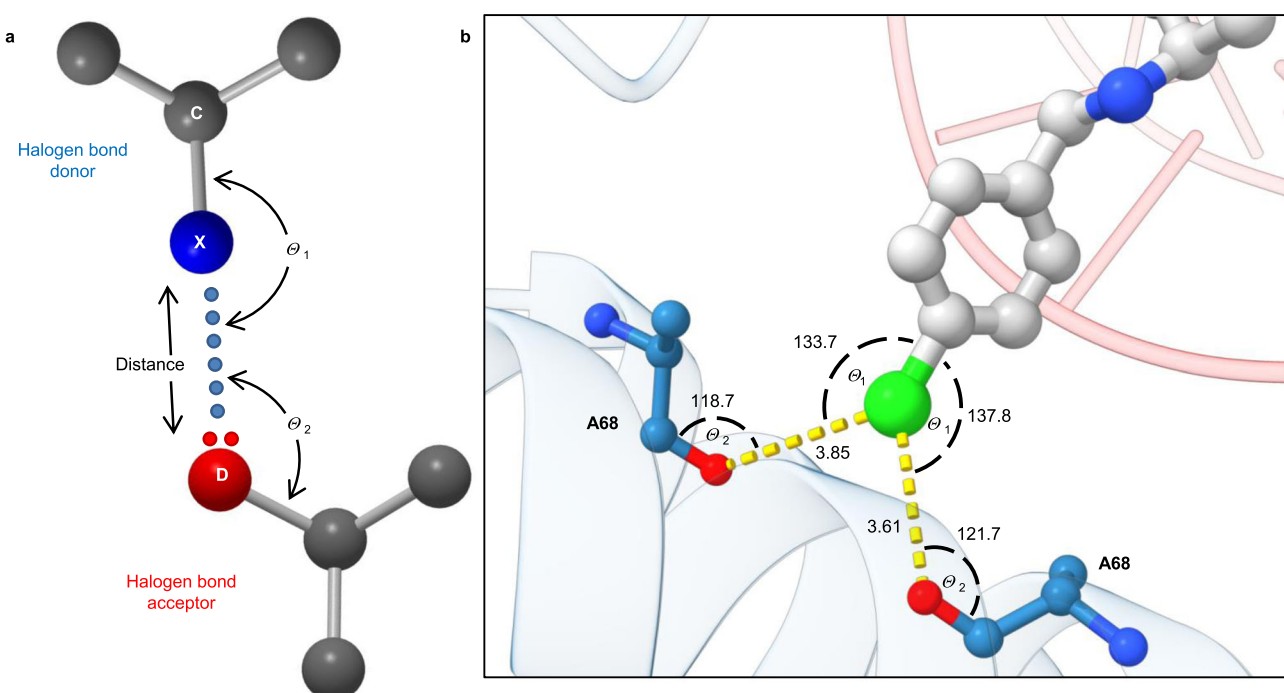

**Fig. 7 Halogen bonding. a** Classical interpretation of halogen bonding involves a linear interaction, as defined by $\theta_1$ and $\theta_2$ angles—especially the $\theta_2$ angle, which depends on the nature of the heteroatom. **b** Halogen bonding in crystal complex of **4** (balls and sticks representation colour coded by element) with *S. aureus* gyrase (represented as cartoon in blue) and oligonucleotide (represented as cartoon in red). Ala68 are in balls and sticks representation colour coded by element.

stacking interaction with a base pair on each side. If each base pair needs to rotate to optimize these stacking interactions, they would do so in the same direction. Therefore, this shift would result in one scissile phosphate getting closer to the catalytic site while the other moves away due to the C2 symmetry in the catalytic sites. We observe a different configuration for the catalytic metal on each side, correlating with the position of the scissile phosphate. This is consistent with the idea that the position of the scissile phosphate determines the configuration of

the catalytic site. Only when the phosphate is close enough to the coordinating acidic residues can a metal be contacted, therefore permitting phosphotransfer. In this case, the metal adopts an A-configuration, consistent with previous obervations[7,10,12] that the catalytic metal establishes contact with Asp508 and Glu435 in addition to the phosphate. When the phosphate is away from the catalytic metal, we observe a B-configuration where the metal contacts residues Asp508 and Asp510, again in line with previous observations[7]. This is consistent with a single metal adopting two

**Table 2 Halogen bond parameters of compound 4 bound to DNA gyrase.**

|  | Ala68 B | Ala68 D |
|---|---|---|
| *Angle* |  |  |
| $\theta_1$ C-X⋯O | 133.7 | 137.8 |
| Threshold: $(140° \leq \theta_1 \leq 180°)$ |  |  |
| $\theta_2$ X⋯O=C | 118.7 | 121.7 |
| Threshold: $(90° \leq \theta_2 \sim 120°)$ |  |  |
| *X⋯D distance* |  |  |
| Cl⋯O | 3.85 | 3.61 |
| Threshold: (<3.27 Å) |  |  |

different configurations, depending on the position of the scissile phosphate, only one of which is catalytic.

There is, however, some evidence for a two-metal mechanism for type IIA topoisomerases[19–21]. On the B-configuration side, there is an evidence for a second metal ion where the A-metal would be expected (Supplementary Fig. 5). An anomalous difference Fourier map shows a peak adjacent to Glu435 that is ~2.1 Å from the carboxyl, consistent with the presence of another metal ion. However, the peak is somewhat weaker than that associated with the B-metal and there is insufficient electron density to model a full coordination sphere. Therefore, it most likely represents a low occupancy site overlapping with a partially occupied water molecule. For simplicity, we opted to model only the latter at full occupancy (Supplementary Fig. 5). It is possible that a second metal could bind some of the time when the phosphate is away. However, we note that, when the phosphate coordinates an A-configuration metal on the other side, no density is observed at the B site. Therefore, the catalytic configuration seems to involve only one metal and we conclude that this structure is consistent with one catalytic metal whose position is influenced by the scissile phosphate. We surmise that this is also relevant in the absence of compound, when the enzyme adopts the various configurations of its catalytic cycle. We speculate that the different conformations adopted by the enzyme could influence the position of the scissile phosphate and therefore when cleavage and resealing occur.

Another important feature of the present crystal structure is that it moves away from the classical paradigm of halogen bonding. The importance of halogen bonding in medicinal chemistry has been recognized as a powerful tool, comparable in some instances to hydrogen bonding, to enhance the binding affinity in drug design[22]. Halogens may be viewed as hydrophobic counterparts of hydrogen-bond donors, and their clear advantage are fewer desolvation costs paid upon forming halogen bond[23]. This is especially true in tight hydrophobic binding sites like the one of DNA gyrase A. The bifurcated symmetrical halogen bonds of [Y–X–Y] type have been described on several occasions in model systems of small molecules in crystal engineering where a single halogen is complexed with two heteroatoms (Y = N, O)[24,25]. Dumele et al. have shown that halogen bonds may exhibit a dominant influence on the small molecule crystal packing thus demonstrating the importance of their enthalpic contribution to the overall binding free enthalpy[26]. To study geometric boundary conditions in bifurcated symmetrical halogen bonds, we have performed an extensive search of small molecules that form bifurcated halogen bonds in Cambridge Structure Database[27] (CSD version 2020.1, data shown in Supplementary Information). Supplementary Tables 9–12 represent information about X⋯O distances and $\theta_1$ and $\theta_2$ angles, while statistics on their geometric parameters are shown in Supplementary Figs. 18–21. The survey data demonstrate that $\theta_1$ and $\theta_2$ angles may occupy a wide range of values (Cl: $70° < \theta_1 < 180°$, $50° < \theta_2 < 170°$; Br: $130° < \theta_1 < 170°$, $90° < \theta_2 < 140°$; I: $130° < \theta_1 < 170°$, $80° < \theta_2 < 150°$),

which are out of the limits defined for a classical halogen bond[16]. The same might be said for the X⋯O distances, which supersede the classical values in the most abundant Cl⋯O cases (Cl⋯O < 3.6 Å, Br⋯O < 3.4 Å, I⋯O < 3.5 Å).

The bifurcated halogen bonds were reported to exist in biological systems in just seven cases (data from PDB, Supplementary Table 8 and Supplementary Fig. 17)[28], and majority incorporate two diverse halogen bond acceptors, like, for example, the backbone carbonyl of Val41 and a sulfur atom of Cys42 in a trypsin-like serine protease protein[29].

In our case, one of the most interesting features of the crystal structure presented here (Fig. 7) is that the chlorine from compound **4** forms a symmetrical bifurcated halogen bond with the backbone carbonyls of both Ala68 residues, which is a consequence of the GyrA interface offering a completely symmetrical binding site. A classical halogen bond is necessarily almost linear in geometry: the $\theta_1$ angle [C-X⋯D] defined by C-X bond (X = halogen) and X⋯D (D = O or N, S) should be close to 180° (Fig. 6a)[17]. $\theta_1$ angle deviations between 25° and 30° correspond to a 50% reduction in halogen bond strength, while deviations >40° are forbidden for halogen bond formation since the σ-hole becomes inaccessible to the electron donor and halogen electron density clashes with the lone electron pair of the heteroatom[30]. Besides the $\theta_1$ angle, a halogen bond strength further depends on the $\theta_2$ [X⋯O=C] angle, the size of the σ-hole on the halogen (influenced by the halogen–donor atom properties), and internuclear distance between halogen and halogen bond acceptor (X⋯D distance). The threshold values of the distance vary but in general is <4 Å. In our case, the $\theta_1$ angle deviates slightly above 40° and the X⋯D distance is slightly suboptimal, which renders the observed halogen interaction less probable by a classical interpretation (Table 2).

Our crystal structure clearly shows that distinct symmetrical bifurcated halogen bond may be obtained in a specific case where two carbonyl oxygens are involved in halogen bonding. Each of two halogen bonds are most probably weaker than a single halogen bond due to suboptimal geometry, but the observed angle and distance values may probably be explained by the sum of interaction energies of a double halogen bonding compared to a single one (Fig. 7b, Supplementary Fig. 8, and Supporting Table 5). The interaction energy of bifurcated halogen bonds can be calculated by density-based as well as orbital-based methods, as shown for small molecules, and is used to study the geometry and the nature of the halogen bonds[31]. In order to get insight to the interactions between compound **4** and the protein, we performed quantum–mechanical calculations of interaction energies on the MP2/aug-cc-pVDZ level of theory[32,33] utilizing Gaussian09 suite of programs[34]. The applied basis set is flexible enough to faithfully describe electron density, while MP2 method provides dispersion energy when calculating molecular interactions. Protein was truncated to Ala68B and Ala68D by adding methyl groups to both Ala68 residues. Moreover, compound **4** was truncated to phenyl ring containing chlorine atom, while the rest of the ligand was replaced by a hydrogen atom (Supplementary Fig. 8). We considered experimental geometries and in this way crystal field effects were implicitly taken into account. Interaction energies were calculated by a method of supermolecules by considering (a) both Ala68 residues, (b) Ala68B only, and (c) Ala68D only. The corresponding interaction energies are −4.23, −2.00, and −2.02 kcal mol$^{-1}$, respectively (Supplementary Table 5). Our calculations demonstrate that the interaction energies are not additive, which gives strong evidence about the role of electron polarization effects. The fact that interaction energies for both Ala68 residues are basically the same despite that carbonyl oxygen atom and chlorine atom distances are slightly different (3.85 and 3.61 Å, respectively) gives some

evidence that both experimental geometrical parameters of the binding site are not perfectly accurate and that there is still space for improvement. Furthermore, these calculations demonstrate that dispersion component of the interaction energy is essential since the Hartree–Fock calculated energy between the compound **4** and both Ala68 residues is positive with the value of 1.38 kcal mol$^{-1}$ (Supplementary Table 5). We offer two additional arguments to corroborate the calculated probability of symmetrical halogen bond: (a) MD simulation snapshots point to the ability of symmetrical bifurcated halogen bonding propensity and (b) such bonding is energetically favourable because the increase in potency of the compounds suggests an important contribution of this symmetrical bifurcated bonding to the improved affinity of the compound. Namely, the IC$_{50}$ values for the *S. aureus* enzyme in the series of our *p*-substituted phenyl derivatives (Table 1) remained basically the same when switching from –H (IC$_{50}$ = 1.02 μM) to –F (IC$_{50}$ = 0.55 μM), as expected since the fluorine substituent is not able to participate in σ-hole bonding. In case of –Cl (IC$_{50}$ = 0.035 μM) and –Br (IC$_{50}$ = 0.007 μM), the difference of approximately one order of magnitude is even bigger than reported for a classical halogen bonding[22], which is probably the consequence of the symmetrical bifurcated halogen bonding. In the case of the iodo substituent (IC$_{50}$ = 0.011 μM), the IC$_{50}$ was comparable to that of bromine. This is probably due to steric hindrance in case of the largest iodo substituent, whose lone electron pairs are likely at the edge of repulsing those of the Ala68 carbonyl oxygens. The same does not happen in the case of a classical single halogen bonding, where an additional increase in potency should be observed for the iodo derivative.

Finally, it is important to note that, to the best of our knowledge, such a so-called symmetrical bifurcated halogen bond has not been identified up until now in a relevant biological system where a ligand interacts with its macromolecular target. This, together with other bifurcated halogen bonds in both CSD and PDB databases, expands the traditional paradigm of halogen bonding and demonstrates that threshold values for X···O bond lengths and $\theta_1/\theta_2$ angles in halogen bonding should be redefined to detect more bifurcated halogen bonds in the crystal structures yet to come. To do so, we propose a new set of boundary conditions for X···O bond lengths (X···O < 4 Å or even more) and $\theta$ angles (70° ≤ $\theta_1$ [C-X···O] ≤ 180° and 60° ≤ $\theta_2$ [X···O=C] ~ 170°).

## Methods

### Molecular docking calculations.
Molecular docking calculations of our series of NBTI compounds within the NBTI binding pocket of *S. aureus* DNA gyrase were performed by the GOLD docking suite in a flexible fashion[35]. Two crystal structures were used for calculations: *S. aureus* DNA gyrase in complex with DNA and an intercalated NBTI ligand (GSK299423 (**1**); PDB ID: 2XCS) and our *S. aureus* DNA gyrase crystal structure with DNA and compound **4**. The experimental coordinates of the co-crystallized NBTI ligand (GSK299423) and compound **4** were used to define the binding site (cavity radius of 16 Å) and all ligands were protonated. The same settings and technical parameters of the GOLD genetic algorithm (population size = 100, selection pressure = 1.1, number of operations = 100.000, number of islands = 5, niche size = 2, migrate = 10, mutate = 95, crossover = 95) were used for all calculations by docking each molecule 10 times into the binding site. Amino acids Met75 harboured on α3 helix, Asp83 on α4 helix, and Met121 were treated as flexible during the docking. The molecular docking protocol was validated by re-docking the co-crystallized NBTI ligands (GSK299423 and compound **4**, respectively) threefold to reproduce their spatial conformation and orientation. The heavy-atom root-mean-square deviation (RMSD ≤ 2.0 Å) between each calculated docking pose and co-crystallized ligand conformation served as decisive criteria for quality of all structure-based settings[36]. The quality of the calculated docked poses was first visually examined in terms of their correct spatial orientation and conformation within the NBTI binding pocket relative to the natively present co-crystallized ligand conformation and additionally by the calculated GOLDScore Fitness function[34]. The best calculated NBTI docked poses in complex with both *S. aureus* gyrase served as initial DNA gyrase–NBTI configurations for performing MD simulations to estimate their stability. For comparative purposes, an apo (ligand-free) structural model of *S. aureus* gyrase (without GSK299423 co-crystallized ligand; PDB ID: 2XCS) was constructed and included in MD simulations as well.

### MD simulations.
All-atom MD simulations were performed utilizing the AMBER18 package[37]. The AMBER-ff14SB[38] and DNA-OL15[39] force fields (FF) were employed for parameterization of the protein and DNA, respectively. Partial atomic charges for each NBTI ligand (**2**–**6**) were calculated using Gaussian09[34] on their geometry-optimized structures by engaging population analysis according to the Merz–Kollman scheme at Hartree–Fock level of theory utilizing 6-31G* basis set for compounds **2**–**5** and a mixed 3-21G/6-31G* basis set for the iodine-containing ligand **6**. The Antechamber module of AMBER18 was employed for obtaining the RESP charges as well as the other ligand FF parameters, utilizing bond lengths and angles obtained from the ligand's optimized geometries[40]. Thus parameterized systems were first neutralized by adding Na$^+$/Cl$^-$ counterions[41] and subsequently immersed in a 10 Å cubic box (128 Å × 130 Å × 112 Å) comprised of TIP3P water molecules[42], resulting in ~176,998 atoms per simulation system.

Prior to performing the simulations, all systems (apo and gyrase-NBTI complexes) were initially subjected to steepest-descent energy minimization to avoid any van der Waals clashes between the atoms as well as to correct poor geometries of the protein side-chain residues. An extensive equilibration comprised of sequential heating on the fully restrained systems from 0 to 150 K for 2 ns and 150–303 K in next 2 ns, followed by additional 10 ns unconstrained NPT equilibration was performed to ensure the integrity of the simulation systems. Production simulations were performed on the canonical (NVT) ensemble using periodic boundary conditions on fully unrestrained systems in total duration of 500 ns per system utilizing a time step of 2 fs. Particle-mesh Ewald method[43] was implemented to account for long-range electrostatic interactions.

The analyses of resulting MD production trajectories (stripped of water and counterions), for each system separately, including RMSD and radius of gyration (Rg), as well as quantification of the halogen bonding propensities, were conducted using the VMD software package[44] as well as cpptraj module of Ambertools 18[37].

Source data from MD simulations is supplied as an additional supplementary file (Supplementary Dataset 5, MD_halo_comp3-6.xlsx).

### Chemistry.
All chemicals were obtained from commercial sources and were used without further purification. The reactions requiring anhydrous conditions were carried out under inert argon atmosphere with anhydrous solvents. Reactions were monitored by thin-layer chromatography on Merck silica gel (60 F254) plates (0.25 mm thick), visualized by ultraviolet light and/or staining reagents. Column chromatography for compound purification of the final compounds was carried out on silica gel 60 (particle size 0.040–0.063 mm; Merck). High-performance liquid chromatography (LC) purity was determined on a Thermo Scientific DIONEX UltiMate 3000 instrument equipped with a diode array detector using an Acquity UPLC® BEH C8 column (1.7 μm, 2.1 mm × 50 mm) for compounds **1**, **2** and **5**–**8**, while Acquity UPLC HSS C18 SB column (1.8 μm, 2.1 mm × 50 mm) was used for compounds **3** and **4**. The solvent system for **1**, **2**, and **5**–**8** consisted of 0.1% trifluoroacetic acid in water (A) and acetonitrile (B), employing the following gradient: 90% A to 50% A in 6 min, then 50% A for 3 min, with a flow rate of 0.3 mL/min and injection volume 5 μL. For **3** and **4**, the solvent system consisted of 0.1% trifluoroacetic acid in water (A) and MeOH (B), employing the following gradient: 95% A to 5% A in 22 min, then 5% A for 2 min, with flow rate 0.3 mL/min and injection volume 5 μL. Melting points were determined on a Reichert hot stage microscope and are uncorrected. NMR spectra ($^1$H and $^{13}$C) were obtained on a Bruker AVANCE III spectrometer (Bruker Corporation. Billerica, MA, USA) at 400 and 100 MHz, respectively, in dimethyl sulfoxide (DMSO)-d$_6$ or CDCl$_3$ solution with tetramethylsilane as an internal standard. High-resolution mass spectra were obtained on Advion CMS (Advion Inc., Ithaca, USA) or VG Analytical Autospec Q mass spectrometer (Fisons, VG Analytical, Manchester, UK). Infrared (IR) spectra were recorded on Thermo Nicolet Nexus 470 ESP FT-IR spectrometer (Thermo Fisher Scientific, Waltham, MA, USA).

### In vitro DNA gyrase inhibitory activity.
A Gyrase Supercoiling High Throughput Plate Assay kit, purchased from Inspiralis (Norwich. UK), was used to determine IC$_{50}$ values of compounds on *S. aureus* and *E. coli* gyrase. The assay was performed on black streptavidin-coated 96-well microtiter plates (Thermo Scientific Pierce), starting by well rehydration with the supplied wash buffer (20 mM Tris-HCl (pH 7.6), 137 mM NaCl, 0.005% (w/v) BSA, 0.05% (v/v) Tween 20). Biotinylated oligonucleotide in wash buffer was immobilized onto the wells, and the excess oligonucleotide was washed off with wash buffer and ultrapure water. In all, 1.5 U of *S. aureus* or *E. coli* gyrase was incubated together with 0.75 μg of relaxed pNO1 plasmid[45] as a substrate in the presence of 3 μL inhibitor solution in 10% DMSO and 0.008% Tween 20 at 37 °C for 30 min in a final reaction volume of 30 μL in buffer (40 mM HEPES·KOH (pH 7.6), 10 mM magnesium acetate, 10 mM dithiothreitol (DTT), 2 mM ATP, 500 mM potassium glutamate, and 0.05 mg/mL albumin for *S. aureus* and 35 mM Tris-HCl (pH 7.5), 24 mM KCl, 4 mM MgCl$_2$, 2 mM DTT, 1.8 mM spermidine, 1 mM ATP, 6.5% (w/v) glycerol, and 0.1 mg/mL albumin for *E. coli*). The reactions were stopped by adding the TF buffer (50 mM NaOAc (pH 4.7), 50 mM NaCl, and 50 mM MgCl$_2$) to allow the triplex formation (biotin–oligonucleotide–plasmid) for an additional 30 min. Afterward, the unbound plasmid was washed off using TF buffer, and the solution of Promega Diamond dye in T10 buffer (10 mM Tris-HCl (pH 8.0), 1 mM EDTA) was added. 10 min after, the solution was mixed and the fluorescence was read using a Tecan Fluorimeter (excitation, 495 nm; emission, 537 nm). Preliminary screening was

performed at four inhibitor concentrations of 100, 10, 1, and 0.1 μM. The $IC_{50}$ values were determined at seven inhibitor concentrations for the compounds that showed residual enzyme activity (<50%) at a concentration of 100 μM, whereas the other compounds were noted as inactive ($IC_{50}$ > 100 μM). $IC_{50}$ was calculated using nonlinear regression-based fitting of inhibition curves using log [inhibitor] versus response-variable slope (four parameters)—symmetrical equation in the GraphPad Prism 6.0 software (GraphPad Software, La Jolla, CA, USA, www.graphpad.com). $IC_{50}$ values represent the mean of two to four independent measurements. Ciprofloxacin was used as a positive control, showing $IC_{50}$s of 93.65 μM and 0.44 μM for *S. aureus* and *E. coli*, respectively.

$IC_{50}$ determination of compounds **5** and **6** was performed by Inspiralis (Norwich, UK) using their gel-based *S. aureus* Gyrase Supercoiling assay (method described in Supplementary information, Supplementary Table 6 and Supplementary Figs. 10–15), due to insufficient sensitivity of fluorescence-based High Throughput Supercoiling assay.

**Human Topo II selectivity evaluation.** A Human topoisomerase II Alpha Relaxation High Throughput Plate Assay kit, purchased from Inspiralis (Norwich, UK), was used to evaluate the selectivity of compounds for the bacterial enzymes over the homologous human topoisomerase II. The assay was performed on black streptavidin-coated 96-well microtiter plates (Thermo Scientific Pierce), starting by wells' rehydration with supplied wash buffer (20 mM Tris-HCl (pH 7.6), 137 mM NaCl, 0.005% (w/v) BSA, 0.05% (v/v) Tween 20). Biotinylated oligonucleotide in wash buffer was immobilized onto the wells and the excess oligonucleotide was washed off with wash buffer. In all, 1.5 U of human topoisomerase IIα enzyme was incubated together with 0.75 μg of supercoiled pNO1 plasmid as a substrate in the presence of 3 μL inhibitor solution in 10% DMSO and 0.008% Tween 20, and 1 μL of 30 mM ATP, at 37 °C for 30 min in a final reaction volume of 30 μL in buffer (50 mM Tris-HCl (pH 7.5), 125 mM NaCl, 10 mM $MgCl_2$, 5 mM DTT, and 100 μg/mL albumin). The reactions were stopped by adding the TF buffer (50 mM NaOAc (pH 4.7), 50 mM NaCl, and 50 mM $MgCl_2$) to allow the triplex formation (biotin–oligonucleotide–plasmid) for an additional 30 min. Afterward, the unbound plasmid was washed off using TF buffer, and the solution of Promega Diamond dye in T10 buffer (10 mM Tris-HCl (pH 8.0), 1 mM EDTA) was added. Ten minutes later, the solution was mixed and the fluorescence was read using Tecan Fluorimeter (excitation, 495 nm; emission, 537 nm). Screening was performed at inhibitor concentrations of 10 μM. Raw data were converted to mean ± SD percentage of residual activity of the enzyme obtained by two independent measurements.

**Antimicrobial susceptibility testing.** Antimicrobial assays were performed by the broth microdilution method in 96-well plate format according to the Clinical and Laboratory Standards Institute guidelines[46] and European Committee on Antimicrobial Susceptibility Testing recommendations[47]. First, bacterial suspensions of specific bacterial strains was prepared to obtain density equivalent to 0.5 McFarland turbidity standard, followed by dilution with cation-adjusted Mueller Hinton broth with TES (Thermo Fisher Scientific), to obtain a final inoculum of $10^5$ colony-forming units/mL. Compounds were dissolved in DMSO and twofold dilutions were prepared, then inoculum was added and the mixture was incubated for 20 h at 37 °C. After incubation, the MIC values were read off by visual assessment as the lowest dilution of compound showing no turbidity. The MICs were determined against *S. aureus* (ATCC 29213), MRSA (NCTC12493), *E. coli* (ATCC 25922), *E. coli* D22, and *E. coli* N43 bacterial strains. Tetracycline was used as a positive control on every assay plate, displaying MICs of 0.5 and 1 μg/mL for *S. aureus* and *E. coli*, respectively.

**Cleavage assays.** Reactions were carried out as described previously[48] with minor adjustments. Five hundred nanograms of relaxed pBR322 plasmid was used as a substrate for each 30 μL reaction. The reaction buffer contained 35 mM Tris pH 7.5, 24 mM KCl, 4 mM $MgCl_2$, 2 mM DTT, and 0.1 mg/mL albumin. Compounds were added at the indicated concentrations and DMSO was present at a concentration of 0.33% (v/v) in all reactions. Reactions were initiated by the addition of 30 nmole of $A_2B_2$ *E. coli* DNA gyrase tetramer and incubated at 37 °C for 30 min. Cleavage complexes were trapped by addition of 7.5 μL of 1% (w/v) SDS. Cleavage products were separated by agarose electrophoresis in the presence 0.5 μg/mL ethidium bromide. In these conditions, intact DNA molecules migrate as a single band as ethidium bromide compacts all topoisomers into a highly supercoiled species. The bands were quantified with the ImageJ software. The proportion of nicked plasmid to the total of each lane was plotted against compound concentration and fitted by the least-square method to the Hill equation using Ipython, matplotlib, numpy, and pandas. The $CC_{50}$ is defined as the concentration of compound producing 50% of the maximum cleavage obtained in the same assay. The corresponding parameter of the best fit was taken as $CC_{50}$.

**Protein cloning, expression, and purification.** The *S. aureus* Gyrase B27-A56 Greek Key deleted Tyr123Phe mutant protein (referred to as SaGyrB27A56-Y123F) was expressed in *E. coli* and purified as described by Bax et al.[10]. For clarity, the expression and purification of the protein is detailed below.

The plasmid pET11- SaGyrB27A56-Y123F was transformed into Ros2plysS cells. These were grown up in 4 L of LB supplemented with carbenicillin and chloramphenicol to OD at 600 nM of approximately 0.6 at 37 °C and then induced with 0.5 mM IPTG at 24 °C for 16 h. The cells were harvested, resuspended in buffer A (Tris·HCl 50 mM, glycerol 10% (v/v), EDTA 1 mM, DTT 1 mM) plus protease inhibitors (Roche Complete, EDTA-free tablets) and lysed in an Avestin cell disruptor. After centrifugation at 30,000 × g for 60 min, the supernatant was loaded onto a 15 mL heparin-sepharose column equilibrated in buffer A and any unbound proteins removed by washing with buffer A. The protein was eluted with a gradient of buffer A with 1 M NaCl. The fractions containing the protein as judged by SDS-polyacrylamide gel electrophoresis were dialyzed against buffer A and loaded onto a 20 mL Q-sepharose column equilibrated in buffer A. After washing with the same buffer to a stable baseline, the protein was eluted with a gradient of buffer A with 1 M NaCl. Fractions containing the protein were pooled and dialyzed against buffer A before loading onto a 10 mL monoQ column equilibrated in buffer A. The protein was eluted gradient of buffer A with 1 M NaCl. Fractions containing the protein were pooled, concentrated, and loaded onto an ENrich SEC 650 size-exclusion column. Fractions containing the protein were pooled, dialyzed against storage buffer (HEPES 20 mM, $Na_2SO_4$ 100 mM, $MnCl_2$ 5 mM, pH 7.0), and concentrated to 12 mg/mL. Aliquots were stored at −80 °C.

DNA oligos (AGCCGTAG and $^P$GTACCTACGGCT) were ordered from Sigma-Aldrich. Each oligo was dissolved in sterile MilliQ water to 2 mM concentration. Equal volumes of each oligo were mixed and annealed in a PCR instrument by heating up the solutions to a starting temperature of 90 °C and gradually decreasing the temperature to 20 °C over 1 h. The 1 mM DNA duplex was stored at −20 °C as 10 μL aliquots.

**Protein crystallization.** Prior to crystallization, 75 μL of the purified of protein at ~12 mg/mL was defrosted and mixed with 36 μL of 20 mM Hepes pH 7.0 and 15 μL of the 1 mM DNA duplex. This gave 126 μL of the protein–DNA complex. All crystallizations were set up using microbatch plates. To set up one microbatch plate, 42 μL of the protein:DNA complex was mixed with 2 μL of 10 mM **4** stock in 100% (v/v) DMSO and an additional 9 μL of buffer 20 mM Hepes, 100 mM NaCl, 5 mM $MnCl_2$, pH 7.0. This protein–DNA–inhibitor complex was incubated for 60 min at room temperature and then centrifuged at 13,000 × g in a benchtop centrifuge for 30 s prior to setting up the crystallization.

Initially a 96-well master plate was made up containing a range of conditions of PEG5000 MME with a range of conditions of bis-tris propane pH 6.2. A microbatch tray (Douglas instruments) was set up by mixing 0.5 μL of the protein–DNA–inhibitor complex with 0.5 μL of the mastermix before covering the drop with paraffin oil. Both the master plate and the microbatch plate was set up using an Oryx8 crystallization robot (Douglas Instruments). Hexagonal crystals appeared after several days from 18% PEG5000 MME (w/v) and 60 mM bis-tris propane pH 6.2. The crystals were harvested into a cryoprotectant solution of 12% (w/v) PEG5000 MME, 140 mM bis-tris propane pH 6.2, 25% (v/v) ethylene glycol, and 0.6 mM **4** before flash-cooling in liquid nitrogen.

**X-ray data collection, processing, and structure solution.** X-ray data were recorded on beamline I04 at the Diamond Light Source (Oxfordshire, UK) using an Eiger2 16M detector (Dectris) with the crystal maintained at 100 K by a Cryojet cryocooler (Oxford Instruments). A total of 3600 images with oscillation of ×0.1° were recorded to a maximum resolution of 2.3 Å. The data were integrated and scaled using DIALS[49] and then merged using AIMLESS[50]. The space group was $P6_1$ with cell parameters of $a = b = 92.6$, $c = 405.5$ Å. Data collection statistics are summarized in Supplementary Table 1.

The majority of the downstream analysis was performed through the CCP4i2 graphical user interface[51]. The structure was solved by direct refinement using REFMAC5[52] of the protein and DNA components taken from a previously solved isomorphous structure of a similar protein–DNA:drug complex (PDB code 5CDP)[53] giving initial $R_{work}$ and $R_{free}$ values of 0.217 and 0.246, respectively. The asymmetric unit contains one copy of the biologically relevant assembly, and at this stage, residual electron density for the **4** was clearly visible at the centre of the complex lying on the non-crystallographic twofold symmetry axis. A ligand restraint dictionary for **4** was prepared using AceDRG[54]. The structure was completed through several iterations of model building with COOT[55] and further restrained refinement with REFMAC5. TLS group definitions obtained from the TLSMD server (http://skuld.bmsc.washington.edu/~tlsmd/)[56] were used in the later stages of refinement. The statistics of the final refined model, including validation output from MolProbity[57], are shown in Supplementary Table 1. Omit $mF_{obs} − DF_{calc}$ difference electron density for the bound ligand was calculated using phases from the final model without the ligand after the application of small random shifts to the atomic coordinates, re-setting temperature factors, and re-refining to convergence. All structural figures were prepared using CCP4mg[58] and ChimeraX[59].

Full PDB X-ray structure validation report is available as an additional supplementary file (Supplementary Dataset 1).

**Reporting summary**. Further information on research design is available in the Nature Research Reporting Summary linked to this article.

## Data availability

Model coordinates and density maps are available in the Protein Data Bank (PDB ID 6Z1A, PDB https://doi.org/10.2210/pdb6Z1A/pdb). The crystal structures used for small molecule database search are publicly available in the Cambridge Structural Database repository (https://www.ccdc.cam.ac.uk/structures/?), using crystal structure identifiers listed in Supplementary Information. The crystal structures used from the pdb database search are publicly available in Protein Data Bank repository (PDB IDs: 2XCS, 5CDP, 1GJD, 1UHI, 4CMJ, 4JYI, 5A86, 5YC6, 5YC7). The source data of Fig. 3 are provided as additional supplementary files (Supplementary Dataset 2 and 3). Other data that support the findings of this study are available from the corresponding author upon reasonable request.

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

## Acknowledgements

The financial support of this work from the Slovenian Research Agency (Grants P1-0017 and P1-0208) is gratefully acknowledged. Work in A.M.'s laboratory is supported by the Biotechnology and Biosciences Research Council (BBSRC; UK) Institute Strategic Programme Grant BB/P012523/1, and the Wellcome Trust (Investigator Award 110072/Z/ 15/Z); J.V.'s work was also supported by an FTMA award from BBSRC (BB/S507921/1). Diamond Light Source is acknowledged for access to beamline I04 under proposal MX18565. The authors would like to thank Professor Dr. Janez Mavri (National Institute of Chemistry, Ljubljana, SI) for all the scientific help and advices on performing QM calculations and Dr. Nicolas Shinada (Discngine S.A.S., Paris, France) for sharing PDB search results.

## Author contributions

A.K., N.M., and M.A. conceived the study. A.K. performed design, chemical reactions, and biological experiments. N.M. performed design. M.H. performed biological experiments. T.G. carried out biochemical assays, and T.G., N.P.B., J.V., C.E.M.S., and D.M.L performed the crystallography and data analysis. N.M., M.A., and A.M. supervised the research. A.K., N.M., and M.A. prepared the manuscript and Supplementary information, with contributions from T.G. and D.M.L. and input from all authors.

## Competing interests

The authors declare no competing interests.
