## [Peer Review File · Nature Communications]

Reviewer #1 (Remarks to the Author):

The authors report on DNA gyrase inhibitors based on a novel design of the LHS and the RHS vectors. The LHS aromatic unit contains a new 1,5-naphthyridyl core and the RHS aromatic needle, that points towards Ala68 into the hydrophobic pocket. A small library of ligands (2–9) with varying *para*-substituent at the RHS aromatic residue is presented along with their synthetic protocol, binding mode, MD simulations, and binding affinities.

The authors have correctly identified that addressing this Ala68 residue in the highly hydrophobic environment could be achieved by halogen bonding interactions. The molecular design is rational, clear, and well-described in manuscript.

The lack of a highly resolved X-ray co-crystal structure with the ligand not being disordered or statistically distributed in two positions adds to the quality of the presented manuscript: here, the first structure with non-disordered ligand binding to this highly interesting target in asymmetric fashion caused by the LHS pocket binding motif is presented. Hence, the novelty is already at very high level and this work will further impact many molecular design strategies on the target.

The synthesis of the target compounds is described with the standards of the field and the structures are established by ^1H NMR spectroscopy, ^{13}C NMR spectroscopy, (sometimes) melting points, *R_f* values, IR spectroscopy, and high-resolution mass spectrometry. The synthetic chemistry is done in well manner.

The central non-covalent interaction discussed in this work is the halogen bonding (XB) of the *para*-substituent (if equal to Cl, Br, I) on the aromatic RHS needle to the backbone carbonyl oxygen atoms of two (symmetric) Ala68. In principle, halogen bonding is well studied in protein–ligand complexation and has been systematically investigated first by the cited reference 21, which should be better highlighted (I am not an author of Ref 21). The authors introduce the rather infamous halogen bond using several references but miss out several key contributions and reviews that could serve the medicinal chemistry community with a better introduction to this interaction (see list below, Ref 1). When the authors analyzed the short contacts of the halogens to the oxygen carbonyl atom of Ala68, a so called symmetric bifurcated halogen bonding motif was identified. This finding is remarkable and indeed deserves a deeper analysis. To the best of the authors knowledge — and I agree with that — this is the first structural evidence of such a bifurcated interaction pattern including a halogen bond in protein–ligand complexation. However, it has been reported multiple times before in small molecules (single-crystal X-ray structures and computationally) and the authors missed to cite these reports. The cited references 22 and 23 are extremely misleading as they both describe a symmetric halogen bond of linear geometry with the structure $\text{N}^+ \cdots \text{I} \cdots \text{N} \text{aryl}$. This is out of context and such a cationic iodonium XB mode is not comparable with the XB situation found in the ligand binding by the authors. I recommend excluding References 22+23 from the manuscript (although they are very valuable papers but not relevant in this context).

Recommended references:

- 1) A timely review on the topic including discussions on halogen bonding modes can be found here: *Angew. Chem. Int. Ed.* 2015, 54, 3290–3327.
- 2) Symmetrical Bifurcated Halogen Bond: Design and Synthesis: *Cryst. Growth Des.* 2011, 11, 3622–3628.
- 3) A single-crystal X-ray of a bifurcated symmetric halogen bond: *Org. Lett.* 2014, 16, 4722–4725.

4) A computational report on “asymmetric bifurcated halogen bonds”: *Phys. Chem. Chem. Phys.* 2015, 17, 6440–6450.

The presence of an attractive XB is supported by the affinity data for the DNA supercoiling inhibition (*S. aureus* and *E. coli*): IC50 values for *E. coli* drop for X=F to X=Cl by a factor of 10, reflecting the repulsive nature of R–F•••O=C interactions and the attractive interaction of the chlorine halogen bond R–Cl•••(O=C)₂. The authors must correct their statement that F engages in weak halogen bonding. This is simply wrong and misleading, see my comments below. The further series of halogens increases binding down to 0.57 μ M for Br and 0.28 μ M for I, in agreement with the expected XB trend Cl < Br < I.

Even more dramatic increases of the affinities are seen for *S. aureus* inhibition where ligand 5 (X = Br) induces a single-digit nanomolar binding (0.007 μ M). This is remarkable given the change in only one single atom and the resulting affinity gain. Changing the *para*-phenyl substituent to the better XB donor X = I (ligand 6) causes a slight affinity drop to 11 nM, presumably caused by the increased steric demand of the larger iodine atom compared to Br and Cl. This is fully expected and well described by the authors. Such steric demands could only be compensated if the ligand's scaffold is flexible in conformation and could potentially adjust the penetration depths into the RHS pocket to target both Ala68 carbonyls.

The crystallography work seems to be done on a reasonable manner, although I cannot judge this technical part. The resolution of 2.3 Å seems to serve as a reasonable basis for discussions on the atomistic level. Especially since little disorder and precise positions of the heavy halogen atoms can be assumed. Another crystallographic expert reviewer should double-check on the applied ligand restraint dictionary AceDRG method and judge the consequences for discussion of this structure.

The calculated MD trajectories of ligands 3–6 are interesting but do not reflect the correct non-covalent interactions. This is seen in the extremely short contact (Table S3) $d_1 = 3.45$ Å for fluoro ligand 2. The C=O•••F interaction must be repulsive (simply by an electrostatic argumentation) and it cannot account for the shortest distance among all halogenated compounds. The authors should review their force fields in the MD simulation and comment on this fact.

One analysis method is clearly missing: A fully updated database search of 1) small molecules in such bifurcated XB complexes. Examples must be shown in the SI and statistics on their geometries should be plotted (histograms of angles, distances and a plot of angles vs. distance). And 2) a PDB databank search must be included, to show potentially other (so far) overseen examples of bifurcated halogen bonds in the biological environment (again mention the results in the main text but provide the full analysis with examples in the SI). If no such bifurcated XB examples can be found in the PDB, that statement should be also mentioned. In any case, the precise databank search criteria must be reported. Such database mining methods are definitely a missing part in the manuscript.

An energetic (DG gain) statement of a single R–X...O=C halogen bond VERSUS the bifurcated R–X...(O=C)₂ halogen bond would be highly appreciated. In the best case, the authors should search for advice in the computation/theoretical literature. Maybe my listed reference 4 (see list above) has some insights? Otherwise, I invite the others to conduct such a theoretical investigation and include it into the manuscript, or publish it separately.

One additional ligand that would be highly interesting in this series is the “methyl” substituted version. It is isosteric to bromine and accounts for roughly the same hydrophobicity as the higher halogens without bearing a sigma-hole for XB. This optional ligand is not mandatory for publication,

but only recommended. Along this line, *log P* values (solubility, water/hexane partitioning values) should be listed along with the binding affinities (also if only calculated). Drug binding is largely affected by the hydrophobicity of ligands and hence it should be at least provided as calculated *log P*.

I suggest to change the nomenclature of the central topic (and title) from “symmetrical halogen bond” to “bifurcated halogen bond”. Please consult my provided references 2–4 (list above) as support for my suggestion. Also “symmetric bifurcated halogen bond” could be an acceptable term.

The section of the conclusions in the manuscript states “This, together with other bifurcated halogen bonds, contradicts the traditional paradigm of halogen bonding”, which I do not support. The statement is too strong. Of course, the “traditional paradigm” that halogen bonding is preferred at linear angles (C–X...acceptor) of 180° is valid but it is not in contrast to the found bifurcated XB, where deviations are valid as well. If medicinal chemists (and docking software) would not rely on the stringent angle requirement but rather on the sigma-hole theory and the exact position of the positivated potential surface at the end cap of halogens R-X, this would not be a traditional paradigm. Hence, I suggest to change the statement and propose that the authors provide short and precise design rules to help the community installing more such bifurcated halogen bonds.

Supporting Information:

1) This statement in the Supporting Information is drastically wrong:

“Such behavior was expected to a certain extent, particularly in the case of (3), since despite the fluorine’s high electronegativity as well as lack of polarizability it can still be involved in halogen bonding interactions, however significantly weaker compared to chloride, bromide, and iodine containing compounds, as previously hypothesized (Fig. S1 and Table S2) (2,3)”

It has NOT been hypothesized that F undergoes weaker halogen bonding. Fluorine does clearly not engage in any kind of halogen bonding. If at all, it undergoes weak hydrogen bonding. In the cited literature (their Ref 2 in the SI), it was found that a fluoroaryl does not undergo halogen bonding but instead the glycine’s carbonyl C=O was bridged by a water to the fluorine atom by hydrogen bonding in the overall pattern (see PDB code 2XU4):

(Gly)C=O...HOH...F—Cl_{ligand}. And this is the reason for the high affinity, because the close contact of C=O to the F atom is repulsive! All other halogens showed halogen bonding interactions following the pattern

(Gly)C=O...X—Cl_{ligand} with X = Cl, Br, I. The X-ray co-crystal structures for the mentioned cases are published here: *ChemMedChem* 2011, 6, 2048–2054. Maybe cite this work as well.

5) Very minor comments on synthetic procedures:

- I do not support reporting such precise yields (one decimal is too much). Please give rounded two-digit numbers as percentages, as in “xy%”.
- Formatting should be reviewed. Many scientifically formal superscripts and subscript characters were neglected, as in: CHCl₃, R_f, Et₃N. Physical quantities must be formatted italics (d, m/z, J). Please use a blank character between number and unit (not 0.63g, but write instead: 0.63 g, etc.). Please insert a unit for the listed NMR chemical shifts (ppm). Remove superfluous blank characters, as in (m/z) for compound 2. Please provide the temperature of the NMR experiments.
- Change the SI headline “Determination of IC50s” to “Determination of IC₅₀ values”

Summary:

The manuscript will have a high impact on the design of active drugs beyond this target. The novelty is high, the structural work is a major step forward on this target, and the consequences for halogen bonding design rules for medicinal chemistry is significant. However, the authors must include the given suggestions, especially 1) the citation of known bifurcated halogen bonds, 2) the correction of the role of fluorine in XB (repulsive!), and 3) the small molecule and protein database searches.

I highly recommend publication of the manuscript in *Nature Communications* after major revisions of my raised points above,

Reviewer #2 (Remarks to the Author):

In the manuscript under consideration, Kolaric and coauthors report on the design and synthesis of a new branch of novel bacterial type II topoisomerase inhibitors (NBTIs) that feature the presence of a phenyl halide moiety. Functionally, the new NBTIs potently inhibit the gyrase-catalyzed DNA supercoiling activity by arresting the gyrase cleavage complex, which, as the authors put it, translates into outstanding antibacterial activities. Crystallographic analysis of gyrase cleavage complex stabilized by compound 4 revealed the bicyclic methoxynaphthyridine ring inserts asymmetrically between +2 and +3 base pairs, which breaks the symmetry otherwise would be expected for the two cleavage centers and likely accounts for the compound's tendency for inducing DNA single-strand break. Also the authors noted the compound's chlorine atom contacts the carbonyl oxygen of Ala68 and its dyad-related mate, likely forming a novel type of bifurcated halogen bond. Overall, this study furthers our understanding on the mechanism of action and structure-activity relationship of NBTIs and suggests the potential use of halogen-derivatized NBTIs in combating bacterial infections. The following suggestions may help in further improving the manuscript.

(1) Crystal structure shows the chlorine atom is 3.8 and 3.6 Å away from the two carbonyl oxygens (Fig. 6), these distances are alarmingly long for them to be classified as halogen bonding. For chlorine-mediated halogen bonding, the chlorine-receptor distances usually fall within 2.8 and 3.2 Å; the bonding energy drops sharply as the distance increases, let alone the angles being sub-optimal. Indeed, the MD simulation carried out by the authors (Fig. 2) placed the chlorine much closer to the accepting oxygens. That said, I'm not sure it's chemically sensible to claim the observed halogen-oxygen interactions as halogen bonding.

(2) The formation of topoisomerase-mediated DNA breaks may result from drug-induced enhancement of cleavage or blockage of religation. Is it possible to distinguish between these two alternatives using the new, disorder-free structure?

(3) The structural effects of methoxy group on the observed asymmetric deformation of DNA should be analyzed in greater detail.

(4) Some NBTIs are currently in late stage clinical trials. So comments from the authors regarding the safety and in vivo efficacy of these new halogen-derivatized NBTIs would be helpful to highlight the significance of this work. Any results from preliminary animal studies?

(4) The comment that “backbone carbonyls are not affected by mutations ... and should be less prone to developing bacterial resistance” (page 6) deserves more careful thoughts. Mutations can alter the conformation and solvent accessibility of main chain atoms, to say that interactions via backbone are immune to mutations appears too wishful.

(5) Fig. 3 requires significant overhaul. The control lane and dosage were not properly labelled on panel a. Gel images of other compounds can also be included in panel a.

(6) Table 1: SD values are only available for human Topo II?

(7) Fig. 5a is not self-explanatory. I find it difficult to extract information from this figure.

(6) Replace “catalytic manganese” by “catalytic metal”. Manganese is not the physiological metal used by topoisomerases.

Reviewer #3 (Remarks to the Author):

General:

First of all, I would like to thank the authors for this very interesting article which is well written and which was a pleasure to read!

Since the phenomenon of bifurcated halogen bonds with a symmetrical partner per se is not novel, the following should be cited: <https://doi.org/10.1016/j.comptc.2014.09.024>.

Furthermore, I have some doubts whether the quality of the structure as reported in this paper is good enough to make the statement of unforeseen symmetrical halogen bond. What about an overlay of two asymmetric halogen bonds concomitantly present in the structure (see comment concerning Line 397)? Are there additional data supporting one or the other scenario (asymmetrical vs. symmetrical)? All this is difficult to judge without pdb. In any case, a more in-depth discussion would be appreciated. In my opinion, besides major revisions needed, whether or not this manuscript may be accepted for publication in this journal largely depends on providing some more/better evidence supporting the claims made, ultimately answering the question: is this just an

exotic but very interesting exception due to peculiarities of this unique ternary complex vs. has the observation made by the authors a broader impact?

Specific Comments and Questions:

Line 98ff

Interesting but overly lengthy explanation and theoretization

Lines 107-109

To the best of my knowledge, there are no data in the manuscript or references made supporting this statement. Please comment as to why this seems to be the case (i.e. demand to investigate e.g. by use of Met121 mutant)

Lines 123-125

The authors of this manuscript have the chance to contribute to the extension on the, still rare, knowledge of halogen bonding, used in medicinal chemistry. I would, therefore, encourage the authors to add slightly more detail to their interpretation of the orbital theory.

As I understood halogen bonding from the cited papers below, I thought that the key point is, that the single p_z orbital electron engaged in a sigma-bond with the aromatic ring and the lack of an electron on the opposed side of the p_z orbital results in the positive sigma hole by exposing the nuclei of the halogen atom.

Cited from: PROTEIN SCIENCE 2013 VOL 22:139—152

»Group VII atoms have five electrons residing in the p-atomic orbitals of the valence shell and that, according to molecular orbital theory, it is the single valence electron of the p_z orbital that participates in forming a covalent r-bond to a carbon atom. Consequently, the depopulation of this orbital opposite the CAX r-bond leaves a hole that partially exposes the positive nuclear charge. This r-hole accounts for the electropositive crown and polar flattening associated with the polarization effects predicted from the QM calculations, whereas the four electrons remaining in the p_x, p_y orbitals account for the electronegative ring lying perpendicular to the r-bond.«

Please also see: Chem. Rev. 2016, 116, 2478–2601, page 2490

»This positive region has been denoted by Politzer et al.¹⁴⁵ as a “ σ -hole” because it can be seen as a local deficit of electron charge opposite a σ -bond.«

Lines 152-153

Could you comment whether the observed electron density could result as the average of these, alternating interactions to only one of the GyrA subunits at a time?

Lines 178-179

While making some basic assumptions, this statement could be generalized for Gram-positive bacteria (*S. aureus*). However, when it comes to Gram-negative bacteria (*E. coli*), target inhibition does usually not translate into MICs. Looking at the values it seems that the compounds are not affected by the Gram-negative cell envelope (ratio MIC/IC50 is largely the same in both Gram-positives and Gram-negatives). Please comment.

Line 187

Check enzyme conc for assays: IC50 should not be below 1/2 of enzyme conc.

Lines 195-196

7 and 8 can't perform 2 H-bonds simultaneously in the required directionalities. Also there seems to be not much space for the carbonyl of the amide in 8? were other NH2 or S-containing moieties, small aromatic heterocycles considered for potential interactions with the two Ala 68 carbonyl groups?

Line 197

How can this be explained? Sterically there seems to be little room for 2 methyl groups?

Line 233

Please provide a reference supporting this statement.

Line 234 (line 221), Fig. 3a

Insufficient labeling:

Conc of compound used?

Lane with no compound

Lane with no enzyme control

Lane showing sizes of linear (cut) DNA

Lane showing size of nicked DNA

Line 262, Fig. 4

In this particular orientation the asymmetry is not very obvious to me. Maybe there is a better orientation to make the point here?

How does the Fo-Fc density look in this region? What are the B-factors of the ligand atoms in comparison with the surrounding protein atoms?

Could it be the case that the observed electron density is actually a result of a static disorder of the lower part of the molecule, similar to what was observed for the previous compounds, with slightly different orientations of the aromatic system and ultimately the Cl so that there is actually only one halogen bond with one GyrA subunit per molecule, but randomly distributed within the crystal?

Line 273

Concerning “asymmetry”: Was it built also in the other orientation into the density?

(or how does the superposition look?) The orientation of the density in the picture 4b could profit from a superpositioning to show why it is significantly different in the structure reported here, compared to all previous ones.

Line 311

How does the quality of the electron density look in this particular region (esp. 5b)?

Line 333-334

To make this statement, data have to be absolutely clear, see comment made for line 234

Line 370

Could that low occupancy come from residual static disorder, or is it really a second Mn that is partially present additionally to the first Mn?

Line 390

In this paper:

<https://doi.org/10.1016/j.comptc.2014.09.024>

A bifurcated halogen bond with a symmetric partner is discussed and the halogen bond is not symmetrical and divided into a primary and secondary interaction. Could it be the case, that the Cl discussed here is either oscillating between the 2 states, or is statically disordered in the crystal?

Line 394-396

Does the quality of the local electron density allow this statement? (see comment above)

Line 397

While intuitively a halogen bond is best at 180°, similar to a hydrogen bond interaction, a survey of the PDB/CSD paints a different picture from experimental data. I find the argument from Scholfield et al. compelling that the increased surface of the halogen at 160-165° leads to the 'best' balance between strongest positive point (180°) and strongest negative point (90°) on the ESP of the halogen.

I would highly encourage the authors to come up with a stronger/more definite statement than »should be close to« either by experimental observation (preferred; can of course come from literature) or from QM calculations.

e.g. »To the best of our understanding.. » or »According to experiments...«, »the current knowledge in the field suggest that..«

Again cited from PROTEIN SCIENCE VOL 22:139—152:

»Geometry of X-bonds: The basic concept of the r-hole makes the X-bond a highly directional interaction, as reflected in the angle of approach of the X-bond acceptor to the halogen relative to the direction of the r-bond (H1, Fig. 1). Surveys of H1 angles for small molecule structures in the Cambridge Database²⁹ as well as biomolecular structures in the PDB⁹ indicate a strong preference for a near linear approach of the acceptor toward the electropositive crown of the r-hole, with a significant drop-off as the acceptor approaches the crossing point between the positive and negative electrostatic potentials (H1 ? 140?).«

Especially:

»The balance between the maximum positive electrostatic potential at H1 ¼ 180° with the increase in available surface area of the halogen atom as H1 approaches 90° accounts for the preference for H1 ? 160°–165°.«

Or from Chem. Rev. 2016, 116, 2478–2601 page 2486, 1.4.1 Directionality: »...scatterplots of intermolecular C–X⋯N interaction versus X⋯N distance (X = I, Br, and Cl). Clearly, short and strong XBs are more directional than the long and weak ones, and by reducing the polarizability of the XB donor, the linearity slightly drops (mean values for the C–X⋯N angle are 171.4° for I, 164.1° for Br, and 154.6° for Cl). This trend is general and has also been observed when XB acceptor sites other than nitrogen are used.«

Lines 414-416

Since QM calculations are within the scope of the authors (looking at the GUASSIAN09 section in the methods) and the group clearly has a nice crystal structure why not run a QM calculation to get the individual parts of interaction energies from the two carbonyls and the halogen?

I would suggest to run a QM calculation based on the obtained crystal structure with 1.) both carbonyls, 2.) carbonyl only from Ala68 B, 3.) only from Ala68 D, (maybe even one calculation without halogen) to gauge how much each carbonyl contributes to the sum of interactions.

This would in my mind add to the manuscript another (strong) argument.

Advice on QM is clearly not needed but authors might find this also interesting:
<https://arxiv.org/pdf/1708.09244.pdf>, Pages 19-25

Also calculations of modeled, asymmetric, more classical bifurcated halogen bonding would be interesting for comparison.

Point-by-point response to Reviewers' comments

We thank Reviewers for their praise of the data and topic, and a very comprehensive review. We have thoroughly contemplated on the remarks and corrected our manuscript according to their suggestions. In most cases we have incorporated the recommendations and believe this has led to a substantially stronger manuscript. The Reviewers' comments and our point-by-point responses are presented as follows.

Author's Response to Reviewer #1

The authors report on DNA gyrase inhibitors based on a novel design of the LHS and the RHS vectors. The LHS aromatic unit contains a new 1,5-naphthyridyl core and the RHS aromatic needle, that points towards Ala68 into the hydrophobic pocket. A small library of ligands (2–9) with varying para-substituent at the RHS aromatic residue is presented along with their synthetic protocol, binding mode, MD simulations, and binding affinities.

The authors have correctly identified that addressing this Ala68 residue in the highly hydrophobic environment could be achieved by halogen bonding interactions. The molecular design is rational, clear, and well-described in manuscript.

The lack of a highly resolved X-ray co-crystal structure with the ligand not being disordered or statistically distributed in two positions adds to the quality of the presented manuscript: here, the first structure with non-disordered ligand binding to this highly interesting target in asymmetric fashion caused by the LHS pocket binding motif is presented. Hence, the novelty is already at very high level and this work will further impact many molecular design strategies on the target.

The synthesis of the target compounds is described with the standards of the field and the structures are established by ¹H NMR spectroscopy, ¹³C NMR spectroscopy, (sometimes) melting points, R_f values, IR spectroscopy, and high-resolution mass spectrometry. The synthetic chemistry is done in well manner.

The central non-covalent interaction discussed in this work is the halogen bonding (XB) of the para-substituent (if equal to Cl, Br, I) on the aromatic RHS needle to the backbone carbonyl oxygen atoms of two (symmetric) Ala68.

1) In principle, halogen bonding is well studied in protein–ligand complexation and has been systematically investigated first by the cited reference 21, which should be better highlighted (I am not an author of Ref 21). The authors introduce the rather infamous halogen bond using several references but miss out several key contributions and reviews that could serve the medicinal chemistry community with a better introduction to this interaction (see list below, Ref 1). When the authors analyzed the short contacts of the halogens to the oxygen carbonyl atom of Ala68, a so called symmetric bifurcated halogen

bonding motif was identified. This finding is remarkable and indeed deserves a deeper analysis. To the best of the authors knowledge — and I agree with that — this is the first structural evidence of such a bifurcated interaction pattern including a halogen bond in protein–ligand complexation. However, it has been reported multiple times before in small molecules (single-crystal X-ray structures and computationally) and the authors missed to cite these reports. The cited references 22 and 23 are extremely misleading as they both describe a symmetric halogen bond of linear geometry with the structure $N^+-I\cdots Naryl$. This is out of context and such a cationic iodonium XB mode is not comparable with the XB situation found in the ligand binding by the authors. I recommend excluding References 22+23 from the manuscript (although they are very valuable papers but not relevant in this context).

Recommended references:

- 1) *A timely review on the topic including discussions on halogen bonding modes can be found here: Angew. Chem. Int. Ed. 2015, 54, 3290–3327.*
- 2) *Symmetrical Bifurcated Halogen Bond: Design and Synthesis: Cryst. Growth Des. 2011, 11, 3622–3628.*
- 3) *A single-crystal X-ray of a bifurcated symmetric halogen bond: Org. Lett. 2014, 16, 4722–4725.*
- 4) *A computational report on “asymmetric bifurcated halogen bonds”: Phys. Chem. Chem. Phys. 2015, 17, 6440–6450.*

We agree that the halogen bond concept, especially as presented in the reference 21 could be better highlighted. As requested, we have omitted previously cited references 22 and 23 as well as the part of the text that describes a symmetric halogen bond of linear geometry with the structure $N^+-I\cdots Naryl$. Instead, we introduced a new part of the discussion, which describes an extensive search of small molecules that form bifurcated halogen bonds in CSD and PDB (data shown in the Supplementary information) and the results and interpretation of these surveys. Furthermore, we have included the data of all 4 suggested references in the main text.

2) *The presence of an attractive XB is supported by the affinity data for the DNA supercoiling inhibition (*S. aureus* and *E. coli*): IC_{50} values for *E. coli* drop for $X=F$ to $X=Cl$ by a factor of 10, reflecting the repulsive nature of $R-F\cdots O=C$ interactions and the attractive interaction of the chlorine halogen bond $R-Cl\cdots(O=C)_2$. The authors must correct their statement that *F* engages in weak halogen bonding. This is simply wrong and misleading, see my comments below. The further series of halogens increases binding down to $0.57\ \mu M$ for *Br* and $0.28\ \mu M$ for *I*, in agreement with the expected XB trend $Cl < Br < I$.*

In the main text, we have correctly stated that fluorine does not form halogen bonds: “Namely, the IC_{50} values for the *S. aureus* enzyme in the series of our *p*-substituted phenyl derivatives (Table 1) remained

basically the same when switching from –H ($IC_{50}=1.02 \mu\text{M}$) to –F ($IC_{50}=0.55 \mu\text{M}$), as expected since the fluorine substituent is not able to participate in σ -hole bonding.”.

Although many authors claim that fluorine in general does not form halogen bonds, we have to point to two papers where fluorine has been shown to form σ -hole and is therefore suitable partner for halogen bonding:

- <https://pubs.rsc.org/en/content/articlelanding/2013/cc/c3cc43513j#!divAbstract>
- <https://pubs.acs.org/doi/10.1021/cg200888n>.

Nevertheless, we agree with the Reviewer #1 that fluorine is generally not considered as a partner in halogen bonding. So, according to the Reviewer #1 suggestion, the text in the Supplementary information has been changed to: *“Such behaviour was expected to a certain extent, particularly in the case of (3), since despite the fluorine’s high electronegativity as well as lack of polarizability it is not involved in halogen bonding, and thus forms significantly weaker interactions compared to chloride, bromide, and iodine containing compounds, as previously hypothesized (Fig. S1 and Table S2).”.* In several instances, the main text was also modified to exclude any possible misunderstanding.

3) Even more dramatic increases of the affinities are seen for S. aureus inhibition where ligand 5 ($X = \text{Br}$) induces a single-digit nanomolar binding ($0.007 \mu\text{M}$). This is remarkable given the change in only one single atom and the resulting affinity gain. Changing the para-phenyl substituent to the better XB donor $X = \text{I}$ (ligand 6) causes a slight affinity drop to 11 nM, presumably cause by the increased steric demand of the larger iodine atom compared to Br and Cl. This is fully expected and well described by the authors. Such steric demands could only be compensated if the ligand’s scaffold is flexible in conformation and could potentially adjust the penetration depths into the RHS pocket to target both Ala68 carbonyls.

We agree completely with the Reviewer #1. The evidence for this notion comes also from MD simulation, where the aromatic system is deformed to allow an optimal halogen bond (see the answer to Reviewer #3 last issue). More flexible RHS part would however include alkyl scaffold, which would in turn probably result in very reactive alkylating agents.

4) The crystallography work seems to be done on a reasonable manner, although I cannot not judge this technical part. The resolution of 2.3 \AA seems to serve as a reasonable basis for discussions on the atomistic level. Especially since little disorder and precise positions of the heavy halogen atoms can be assumed. Another crystallographic expert reviewer should doublecheck on the applied ligand restraint dictionary AceDRG method and judge the consequences for discussion of this structure.

This issue was tackled in the Supplementary information. Namely, the new Fig. S6 describes crystallographic evidence supporting the placement, orientation and conformation of compound **4** in the complex with DNA gyrase and DNA.

4) The calculated MD trajectories of ligands 3–6 are interesting but do not reflect the correct non-covalent interactions. This is seen in the extremely short contact (Table S3) $d1 = 3.45 \text{ \AA}$ for fluoro ligand 2. The $C=O \cdots F$ interaction must be repulsive (simply by an electrostatic argumentation) and it cannot account for the shortest distance among all halogenated compounds. The authors should review their force fields in the MD simulation and comment on this fact.

We have used General AMBER Force Field (GAFF) that is in our opinion appropriate and accurate enough for parameterization of the present ligands. We believe that the “extremely short contact” between fluoro substituent for ligand **3** is not that extreme and is comparable to distances between other halogen atoms and Ala68 oxygens. What can be deduced from the crystal structure and MD simulation is that very little space is available in the GyrA dimer interface and that *p*-halogenophenyl takes all the space available regardless the halogen involved. Of course, Reviewer #1 is right and the close contact of fluorine does probably cause repulsion while forming no halogen bond thus influencing the potency of compound **3**, which is much lower than that of other halogenated derivatives. On the other hand, comparison of IC_{50} s for H(**2**) and F(**3**) derivatives show a slight increase in inhibitory activity/affinity for fluoro- derivative (**3**) meaning that other interactions take place compensating the electrostatic repulsion. Namely, if we take a closer look of the crystal structure, intense anchoring of the chlorine atom of the compound **4** bound to DNA Gyrase A dimer occurs, which probably involves other-than-halogen bond interactions (see figure):

Crystal structure reveals multiple interactions of the chlorine substituent in **4**: A68 carbonyl oxygens form the symmetrical bifurcated halogen bond, both M75 offer a possible S-C-H...Cl weak electrostatic interactions (dipole-dipole), while both M121 are probably too distant for interactions with -Cl and are more likely in contact with the phenyl. All these interactions (+ vdW interactions) exist in the case of F- substituent as well and probably compensate the repulsion, which at the end leads to the similar poses, but lower affinity compared to -Cl.

5) One analysis method is clearly missing: A fully updated database search of 1) small molecules in such bifurcated XB complexes. Examples must be shown in the SI and statistics on their geometries should be plotted (histograms of angles, distances and a plot of angles vs. distance). And 2) a PDB databank search must be included, to show potentially other (so far) overseen examples of bifurcated halogen bonds in the biological environment (again mention the results in the main text but provide the full analysis with examples in the SI). If no such bifurcated XB examples can be found in the PDB, that statement should be also mentioned. In any case, the precise databank search criteria must be reported. Such database mining methods are definitely a missing part in the manuscript.

We thank the reviewer for the clever suggestion on deeper database search. We have performed the search of small molecules in Cambridge Structure Database (CSD version 2020.1) and in PDB databank, able to form bifurcated halogen bonds. A short discussion on the survey is included in the main text, while the precise databank search criteria and examples for all halogen atoms are exemplified in Tables

S8-11 along with information about X•••O distances, Θ_1 , and Θ_2 angles. Statistics on their geometric parameters are shown on Figures S12-15. While CSD survey detected multiple bifurcated halogen bonds (even symmetrical ones), The PDB survey detected only a few such bonds as presented in this manuscript.

6) An energetic (ΔG gain) statement of a single R–X•••O=C halogen bond VERSUS the bifurcated R–X•••(O=C)₂ halogen bond would be highly appreciated. In the best case, the authors should search for advice in the computation/theoretical literature. Maybe my listed reference 4 (see list above) has some insights? Otherwise, I invite the others to conduct such a theoretical investigation and include it into the manuscript, or publish it separately.

We have performed a theoretical study by quantum-mechanical (QM) calculations of interaction energies considering a) both Ala residues, b) Ala68B only, and c) Ala68D only, and expanded the discussion to include a theoretical evidence that bifurcated halogen bond is energetically favourable compared to each individual halogen bond. Furthermore, we have extended the discussion by incorporating the listed reference 4.

7) One additional ligand that would be highly interesting in this series is the “methyl” substituted version. It is isosteric to bromine and accounts for roughly the same hydrophobicity as the higher halogens without bearing a sigma-hole for XB. This optional ligand is not mandatory for publication, but only recommended.

We thank the reviewer for the suggestion. Since the methyl analogue is not expected to form any of the important interactions (halogen or H-bonds) in the GyrA binding site, we believe that it would not add to the quality of the paper. However, we are preparing other more Medicinal Chemistry oriented paper where we will gladly take this advice and extend the SAR of the current inhibitors' series.

8) Along this line, log P values (solubility, water/hexane partitioning values) should be listed along with the binding affinities (also if only calculated). Drug binding is largely affected by the hydrophobicity of ligands and hence it should be at least provided as calculated log P.

LogP values were calculated and are listed in the Table S4. (logP and logD of compounds 2-9 calculated with MarvinSketch 20.17).

9) I suggest to change the nomenclature of the central topic (and title) from “symmetrical halogen bond” to “bifurcated halogen bond”. Please consult my provided references 2–4 (list above) as support for my suggestion. Also “symmetric bifurcated halogen bond” could be an acceptable term.

We completely agree with the Referee 1. Since not all bifurcated halogen bonds are symmetrical, we prefer to use the term “symmetrical bifurcated halogen bond”.

9) The section of the conclusions in the manuscript states “This, together with other bifurcated halogen bonds, contradicts the traditional paradigm of halogen bonding”, which I do not support. The statement is too strong. Of course, the “traditional paradigm” that halogen bonding is preferred at linear angles (C–X⋯acceptor) of 180° is valid but it is not in contrast to the found bifurcated XB, where deviations are valid as well. If medicinal chemists (and docking software) would not rely on the stringent angle requirement but rather on the sigma-hole theory and the exact position of the positivated potential surface at the end cap of halogens R-X, this would not be a traditional paradigm. Hence, I suggest to change the statement and propose that the authors provide short and precise design rules to help the community installing more such bifurcated halogen bonds.

What we had in mind with our statement is that if the crystal structure is probed with Ligand Scout software for halogen bonding using classical boundary conditions (X⋯O bond lengths (Cl⋯O < 3.27 Å, Br⋯O < 3.37 Å, I⋯O < 3.50 Å) and angles ($140^\circ \leq \Theta_1 [\text{C-X}\cdots\text{O}] \leq 180^\circ$ and $90^\circ \leq \Theta_2 [\text{X}\cdots\text{O}=\text{C}] \sim 120^\circ$; Sirimulla, S., Bailey, J. B., Vegesna, R., Narayan, M. Halogen Interactions in Protein-Ligand Complexes: Implications of Halogen Bonding for Rational Drug Design. *J. Chem. Inf. Model.* **53**, 2781-2791 (2013)), no such bond is detected, as seen in the figure below:

Halogen Bonding

Figure 12.4. Halogen Bonding settings

Halogen Bond Ranges	Effect
Bond Angle ranges	(default: 140.00°)
Acceptor Angle ranges	Calculate the weighted vector from the halogen neighbor atom to all of its neighbors. This mixed direction vector is inverted and checked against the vector (halogen atom, halogen neighbor atom). This angle must be below the specified threshold. (default: 90.00°)
Distance tolerance (VdW corrected)	The distance between the halogen atom and the halogen neighbor must be below the specified threshold. (default: 0.300 Å)
Methionine exclusion angle	(default: 10.00°)

It is true though that most of new publications like Shinada et al. describe the existence of bifurcated halogen bonds (Shinada, N. K., de Brevern, A. G., Schmidtke, P. Halogens in Protein-Ligand Binding Mechanism: A Structural Perspective. *J. Med. Chem.* **62**, 9341-9356 (2019)), but none proposed a new set of boundary conditions. In agreement with Reviewer #1 proposal, we have modified the statement and have included the new boundary conditions that may help the scientific community to identify bifurcated halogen bonds even more easily:

“This, together with other bifurcated halogen bonds in both CSD and PDB databases, expands the traditional paradigm of halogen bonding and demonstrates that threshold values for X···O bond lengths and Θ_1/Θ_2 angles in halogen bonding should be redefined to detect more bifurcated halogen bonds in the crystal structures yet to come. To do so, we propose a new set of boundary conditions for X···O bond lengths ($X\cdots O < 4 \text{ \AA}$ or even more) and Θ angles ($70^\circ \leq \Theta_1 [C-X\cdots O] \leq 180^\circ$ and $60^\circ \leq \Theta_2 [X\cdots O=C] \sim 170^\circ$).

10) Supporting Information:

This statement in the Supporting Information is drastically wrong:

“Such behavior was expected to a certain extent, particularly in the case of (3), since despite the fluorine’s high electronegativity as well as lack of polarizability it can still be involved in halogen bonding interactions, however significantly weaker compared to chloride, bromide, and iodine containing compounds, as previously hypothesized (Fig. S1 and Table S2) (2,3)”

It has NOT been hypothesized that F undergoes weaker halogen bonding. Fluorine does clearly not engage in any kind of halogen bonding. If at all, it undergoes weak hydrogen bonding. In the cited literature (their Ref 2 in the SI), it was found that a fluoroaryl does not undergo halogen bonding but instead the glycine’s carbonyl C=O was bridged by a water to the fluorine atom by hydrogen bonding in the overall pattern (see PDB code 2XU4):

(Gly)C=O...HOH...F—Cligand. And this is the reason for the high affinity, because the close contact of C=O to the F atom is repulsive! All other halogens showed halogen bonding interactions following the pattern

(Gly)C=O...X—Cligand with X = Cl, Br, I. The X-ray co-crystal structures for the mentioned cases are published here: *ChemMedChem* 2011, 6, 2048–2054. Maybe cite this work as well.

As mentioned before, the statement was modified according to the Reviewer’s suggestion.

11) Very minor comments on synthetic procedures:

- I do not support reporting such precise yields (one decimal is too much). Please give rounded two-digit numbers as percentages, as in “xy%”.
- Formatting should be reviewed. Many scientifically formal superscripts and subscript characters were neglected, as in: CHCl₃, Rf, Et₃N. Physical quantities must be formatted italics (δ , m/z, J). Please use a blank character between number and unit (not 0.63g, but write instead: 0.63 g, etc.). Please insert a unit for the listed NMR chemical shifts (ppm). Remove superfluous blank characters, as in (m/z) for compound 2.
- Please provide the temperature of the NMR experiments.
- Change the SI headline “Determination of IC50s” to “Determination of IC50 values”

All the minor issues were modified as suggested by the Reviewer #1.

Summary:

The manuscript will have a high impact on the design of active drugs beyond this target. The novelty is high, the structural work is a major step forward on this target, and the consequences for halogen bonding design rules for medicinal chemistry is significant. However, the authors must include the given suggestions, especially 1) the citation of known bifurcated halogen bonds, 2) the correction of the role of fluorine in XB (repulsive!), and 3) the small molecule and protein database searches.

I highly recommend publication of the manuscript in Nature Communications after major revisions of my raised points above,

Author's Response to Reviewer #2

In the manuscript under consideration, Kolaric and coauthors report on the design and synthesis of a new branch of novel bacterial type II topoisomerase inhibitors (NBTIs) that feature the presence of a phenyl halide moiety. Functionally, the new NBTIs potently inhibit the gyrase-catalyzed DNA supercoiling activity by arresting the gyrase cleavage complex, which, as the authors put it, translates into outstanding antibacterial activities. Crystallographic analysis of gyrase cleavage complex stabilized by compound 4 revealed the bicyclic methoxynaphthyridine ring inserts asymmetrically between +2 and +3 base pairs, which breaks the symmetry otherwise would be expected for the two cleavage centers and likely accounts for the compound's tendency for inducing DNA single-strand break. Also the authors noted the compound's chlorine atom contacts the carbonyl oxygen of Ala68 and its dyad-related mate, likely forming a novel type of bifurcated halogen bond. Overall, this study furthers our understanding on the mechanism of action and structure-activity relationship of NBTIs and suggests the potential use of halogen-derivatized NBTIs in combating bacterial infections. The following suggestions may help in further improving the manuscript.

1) Crystal structure shows the chlorine atom is 3.8 and 3.6 Å away from the two carbonyl oxygens (Fig. 6), these distances are alarmingly long for them to be classified as halogen bonding. For chlorine-mediated halogen bonding, the chlorine-receptor distances usually fall within 2.8 and 3.2 Å; the bonding energy drops sharply as the distance increases, let alone the angles being sub-optimal. Indeed, the MD simulation carried out by the authors (Fig. 2) placed the chlorine much closer to the accepting oxygens. That said, I'm not sure it's chemically sensible to claim the observed halogen-oxygen interactions as halogen bonding.

As discussed before, we have performed the search of small molecules in Cambridge Structure Database (CSD version 2020.1) and in PDB databank, able to form bifurcated halogen bonds. The data clearly show that distances in bifurcated halogen bonds differ substantially in the case of bifurcated bonds. To corroborate our results, we have performed the QM calculations, which prove energetically favourable enthalpic contribution of the bifurcated halogen bond.

2) The formation of topoisomerase-mediated DNA breaks may result from drug-induced enhancement of cleavage or blockage of religation. Is it possible to distinguish between these two alternatives using the new, disorder-free structure?

This is an interesting question, however answering it unambiguously would require a structure where the DNA had been cleaved by the enzyme. In the present structure, despite being highly informative, a doubly nicked DNA is used for crystallisation. Our structure unambiguously points towards the compound influencing the position of the scissile phosphate, which influences its ability to form the catalytic configuration by contacting the catalytic metal. It is likely that such contact is required for both

cleavage and re-ligation. Therefore, our structure indicates that a compound favouring cleavage would move the phosphate closer to the catalytic metal, whereas impairing re-ligation would involve taking it away from the catalytic metal. Consistently, in our structure the scissile phosphate contacts the metal on one side (A-configuration) and not the other (B-configuration). However, since the DNA is already doubly nicked, it is not possible to ascertain which side is cleaved in the cleavage complex. That being said, we favour the interpretation that it is the re-ligation that is impaired, for the following reasons: 1) Since favouring cleavage involves taking the phosphate in closer proximity with the metal, this would also favour re-ligation and the cleavage complex would therefore be unstable. By the same token, taking the phosphate away would hinder re-ligation and cleavage, but cleavage established more slowly can still be “trapped” because re-ligation is impaired. Consistently, in the case of fluoroquinolones, cleavage establishment is quite slow (10-15 minutes), indeed much slower than the catalytic turnover rate. 2) Again in the case of fluoroquinolones, structures of the cleavage complex where the DNA is cleaved by the enzyme show the phosphate away from the catalytic metal when the DNA is cleaved, suggesting impairment of re-ligation. The metal is in the B-configuration, similar to the configuration on one side of our structure. We therefore surmise, by comparison with fluoroquinolones, that the “B-configuration” side would be the cleaved one and that the compounds likely impair re-ligation on one side. However, since in our structure the DNA is already cleaved on both sides (not by the enzyme), it is not possible to know for sure.

3) The structural effects of methoxy group on the observed asymmetric deformation of DNA should be analyzed in greater detail.

According to Reviewer #2 suggestion, we have added a thorough discussion in the main text together with the new Figure 6 that clearly demonstrates how a methoxy group influences the asymmetry of the cleaved DNA molecule.

4) Some NBTIs are currently in late stage clinical trials. So comments from the authors regarding the safety and in vivo efficacy of these new halogen-derivatized NBTIs would be helpful to highlight the significance of this work. Any results from preliminary animal studies?

We are completely aware of that issue, as we have pointed in our latest review article (Kolarič et al. Two-decades of successful SAR-grounded stories of the novel bacterial topoisomerase inhibitors (NBTIs). *J. Med. Chem.* In press, 2020). In the future, we plan to develop new potent NBTI's that will be assayed in *in vivo* neutropenic mouse thigh infection model and *in vivo* safety studies. There are however two important reasons why we still haven't performed these studies and incorporated them in the present manuscript.

1) The first and most important reason is that this paper novelty is in full elucidation of NBTI's mode of action, which was achieved by the first crystal structure without the apparent static disorder and with a clear position of catalytic metal ions. Further novelty is the experimental evidence for the bifurcated symmetrical halogen bonds in a biological setup. Since these results point to more basic science (e. g. Chemical Biology story), we believe that incorporating some more advanced Medicinal Chemistry results would dilute the focus of the manuscript.

2) The second reason is that our compounds have a molecular probe/hit status. We have included Table S6 in the Supplementary information with new results of cytotoxicity studies performed on human HUVEC and HepG2 cell lines and inhibition studies on hERG potassium channels. Since the current results on hERGs were not promising (halogenated compounds have hERG inhibitory activity in low micromolar range), we believe that *in vivo* studies should be done only with new optimized compounds with less or no hERG activity, and we see no point in performing *in vivo* safety studies with compounds that obviously have safety issues. Saying that, our compounds are very good molecular probes and are completely useful for the studies performed and described in the manuscript, but are not yet lead compounds suitable for the *in vivo* experiments.

To clearly point to toxicity issues, we have inserted the next text into the manuscript:

»We have performed preliminary cytotoxicity studies on human HUVEC and HepG2 cell lines and determined hERG potassium channel inhibition (Supporting information Table S6). The results show safety issues related primarily with hERGs inhibition, which is a class-related problem for NBTIs.5. Having this in mind, our compounds should be regarded as functional molecular probes for investigating the mechanism of DNA Gyrase inhibition, while hit-to-lead optimization will be done to expand the current NBTI library and yield candidates with less toxicity issues suitable for in vivo studies.».

5) *The comment that “backbone carbonyls are not affected by mutations ... and should be less prone to developing bacterial resistance” (page 6) deserves more careful thoughts. Mutations can alter the conformation and solvent accessibility of main chain atoms, to say that interactions via backbone are immune to mutations appears too wishful.*

We have therefore re-phrased the sentence as “... Furthermore, the backbone carbonyls cannot be removed by simple mutations and the NBTIs forming interactions with these carbonyls should be less prone to the development of bacterial resistance through target mutation. ...” (please see our comment on Reviewer #3 points as well).

6) Fig. 3 requires significant overhaul. The control lane and dosage were not properly labelled on panel a. Gel images of other compounds can also be included in panel a.

Figure 3 has been re-done. Additionally, Fig. S10 was included in the Supplementary information for more clarity.

7) Table 1: SD values are only available for human Topo II?

In the revised version of the manuscript we have included SD values for all results of enzyme inhibition assays.

8) Fig. 5a is not self-explanatory. I find it difficult to extract information from this figure.

Figure 5a aims to show that the phosphates in +1, +2, +3 and +4 are shifted between the two sides. We have modified the Figure 5 to make this clearer. The two superposed sides are now in different rendering, with extra labels and explanation.

9) Replace “catalytic manganese” by “catalytic metal”. Manganese is not the physiological metal used by topoisomerases.

We have corrected the term according to the Reviewer’s suggestion. However, it is important to note that manganese can replace magnesium and sustain the full catalytic cycle of DNA gyrase. In the crystal structure it is a manganese that is used by the enzyme. It is therefore correct to state “catalytic manganese” at least *in vitro* and *in structuro*. Therefore, at page 14, we propose the statement: “.catalytic metal, in this case a manganese...”. In all other cases “manganese” was changed with “metal”.

Author's Response to Reviewer #3

General:

First of all, I would like to thank the authors for this very interesting article which is well written and which was a pleasure to read!

We thank the Reviewer #3 for a very encouraging comment.

1) Since the phenomenon of bifurcated halogen bonds with a symmetrical partner per se is not novel, the following should be cited: <https://doi.org/10.1016/j.comptc.2014.09.024>.

We are thankful for the suggestion and we have included the new reference. It is true that symmetrical bifurcated halogen bonds (SBXB) are not new, and we have stated that correctly in the submitted version of the manuscript (previous version – references 22 and 23, which were considered by the Reviewer #1 as non relevant, for the current version of the manuscript, please see our CSD study and the reference 28 in the main text). All published SBXB have been observed on small molecules in crystal engineering with the exemption of PDB structures 5YC6 and 5YC7, where however the symmetrical partner is a carboxylate. Since we do not want to overemphasize the importance of the “symmetrical”, we have omitted the word “unforeseen” from the title and have toned down the emphasis of the “symmetrical”, as this is important, but not the main message of the manuscript.

2) Furthermore, I have some doubts whether the quality of the structure as reported in this paper is good enough to make the statement of unforeseen symmetrical halogen bond. What about an overlay of two asymmetric halogen bonds concomitantly present in the structure (see comment concerning Line 397)? Are there additional data supporting one or the other scenario (asymmetrical vs. symmetrical)? All this is difficult to judge without pdb. In any case, a more in-depth discussion would be appreciated. In my opinion, besides major revisions needed, whether or not this manuscript may be accepted for publication in this journal largely depends on providing some more/better evidence supporting the claims made, ultimately answering the question: is this just an exotic but very interesting exception due to peculiarities of this unique ternary complex vs. has the observation made by the authors a broader impact?

We provide a lengthy explanation here as this will simultaneously address several of the subsequent comments and questions. All references to “density” herein refer to the mFobs-DFcalc omit difference density map for the ligand, which is free from bias due to the modelled ligand (see description in crystallography methods).

Inspection of the omit difference density map (Fig. 4b and Fig. S6a) reveals that we have correctly placed the three ring systems. We are especially confident in the placement of the upper methoxy-naphthridine and the lower chlorophenyl moieties as this is where the density is strongest. This will be

because these groups are tethered through interactions with the DNA and the protein, respectively. Indeed, when the omit map is contoured such that only the most electron dense regions are visible, it is clear that the position of the chlorine atom is very well defined (Fig. S6a). The density is weaker for the aminopiperidine central unit, which is understandable because it interacts with neither the protein nor the DNA and is likely to be more flexible as a result. The quality of the density is reflected in the refined B-factors; the overall B-factor for the ligand is somewhat elevated relative to the surrounding protein, but is comparable to that of the DNA. A detailed analysis of the B-factors for the three ring systems (Fig. S6b), shows a noticeably higher value for the aminopiperidine linker, but this is in line with the lack of interactions with protein or DNA in this region of the ligand as discussed above.

The density associated with each ring is flattened (Fig. S6b), thus there are only two potential orientations (or a mixture of two orientations) for the asymmetric methoxy-naphthiridine and the aminopiperidine rings, and only one for the symmetrical chlorophenyl ring. Nonetheless, the distinct asymmetry of the methoxy-naphthiridine ring and of its attachment to the aminopiperidine linker, makes the assignment of its orientation unambiguous. Conversely, the density alone does not define the correct orientation of the aminopiperidine linker (or whether there is a mixture of two orientations). However, only one of these orientations enables the formation of a hydrogen bond to the side-chain of one of the Asp83 residues (Fig. 4a and Fig. S6b), which is donated by the secondary amine linking the aminopiperidine and chlorophenyl rings. This latter orientation is the one chosen for our model. Despite this, it is possible to refine the ligand with the aminopiperidine ring flipped, which gives comparable statistics for the resultant model. Indeed, the coordinates of the methoxy-naphthiridine and chlorophenyl moieties for the two models are closely superposable and the chlorine positions differ by $<0.1 \text{ \AA}$. This prompted us to experiment with refining the two conformations of the ligand simultaneously, each with half occupancy. However, the alternate chlorine positions had a tendency to drift apart a little, but not necessarily towards the carbonyl groups of the Ala68 residues. The shortest chlorine-acceptor distance we achieved in these refinements was 3.36 \AA , although the alternate chlorine position was over 4 \AA distant from the symmetry equivalent Ala68. In addition, the chlorines had moved to positions either side of the peak of the omit density. If this model were representative of the true situation, we would expect the density peak to be elongated along the vector connecting the two chlorine positions, which it was not. Finally, we attempted a further round of refinements with the chlorine-acceptor distances restrained to 3.2 \AA in REFMAC5, this approximating to the upper limit observed elsewhere for halogen bonds. We tested situations where the chlorine in a single fully-occupied conformation was simultaneously restrained to both carbonyls, and situations where a dual occupancy ligand had the alternate chlorine positions restrained to opposing carbonyls. As expected in all such refinements, shorter chlorine-acceptor distances were achieved, but in every case the chlorines were pulled out of the omit difference density peak. Thus, we conclude that a single fully-occupied conformation refined without chlorine-acceptor restraints is most likely the best interpretation of the data. Given that only one

of the two possible orientations of the aminopiperidine enables hydrogen bonding to one of the Asp83 residues, we have chosen this as the most likely ligand conformation.

Specific Comments and Questions:

3) Line 98ff

Interesting but overly lengthy explanation and theoretization

Both Ala68 residues proved to be crucial for the design of our compounds, so we are not sure how to shorten this part of the text without losing the emphasis on this specific amino acid residue.

4) Lines 107-109

To the best of my knowledge, there are no data in the manuscript or references made supporting this statement. Please comment as to why this seems to be the case (i.e. demand to investigate e.g. by use of Met121 mutant)

We believe that there is no need to investigate the statement by using Met121 or other mutants, as we wanted to point to the backbone carbonyls, which remain the same whatever mutant is used. We have therefore re-phrased the sentence as "... Furthermore, the backbone carbonyls cannot be removed by simple mutations and the NBTIs forming interactions with these carbonyls should be less prone to the development of bacterial resistance through target mutation. ..."

5) Lines 123-125

The authors of this manuscript have the chance to contribute to the extension on the, still rare, knowledge of halogen bonding, used in medicinal chemistry. I would, therefore, encourage the authors to add slightly more detail to their interpretation of the orbital theory.

As I understood halogen bonding from the cited papers below, I thought that the key point is, that the single pz orbital electron engaged in a sigma-bond with the aromatic ring and the lack of an electron on the opposed side of the pz orbital results in the positive sigma hole by exposing the nuclei of the halogen atom.

Cited from: PROTEIN SCIENCE 2013 VOL 22:139—152

»Group VII atoms have five electrons residing in the p-atomic orbitals of the valence shell and that, according to molecular orbital theory, it is the single valence electron of the pz orbital that participates in forming a covalent r-bond to a carbon atom. Consequently, the depopulation of this orbital opposite the CAX r-bond leaves a hole that partially exposes the positive nuclear charge. This r-hole accounts for the electropositive crown and polar flattening associated with the polarization effects predicted from

the QM calculations, whereas the four electrons remaining in the p_x , p_y orbitals account for the electronegative ring lying perpendicular to the r -bond.«

Please also see: *Chem. Rev.* 2016, 116, 2478–2601, page 2490

»This positive region has been denoted by Politzer et al.¹⁴⁵ as a “ σ -hole” because it can be seen as a local deficit of electron charge opposite a σ -bond.«

We thank the reviewer for a very clear description of the σ -hole nature. We believe that the manuscript already explains the same, and we cite a part of the Results section, page 7: “Halogen bonding, a non-covalent interaction of halogen atoms, is explained by the presence of a region of positive electrostatic potential, the so-called σ -hole, on the outermost portion of the halogen’s surface, centered on the $R-X$ axis (X =halogen, R =alkyl or aryl carbon). According to Clark et al., in molecules that contain Cl, Br and I atoms, the halogen atoms closely approximate the $s^2p^2 xp^2 yp^1 z$ configuration, where the z -axis is along the $R-X$ bond^{14,15}. The halogen’s three unshared electrons pairs produce a belt of negative electrostatic potential around the central part of X , leaving the outermost region positive, the σ -hole. The σ -hole differs in size according to the halogen involved ($I > Br > Cl > > F$) and offers a possibility for an interaction with a Lewis base, e.g. a lone electron pair of a heteroatom like a carbonyl oxygen.”

6) Lines 152-153

Could you comment whether the observed electron density could result as the average of these, alternating interactions to only one of the GyrA subunits at a time?

Please see response to reviewer 3’s first comment where a very thorough answer to the electron density issue is given.

7) Lines 178-179

While making some basic assumptions, this statement could be generalized for Gram-positive bacteria (*S. aureus*). However, when it comes to Gram-negative bacteria (*E. coli*), target inhibition does usually not translate into MICs. Looking at the values it seems that the compounds are not affected by the Gram-negative cell envelope (ratio MIC/IC50 is largely the same in both Gram-positives and Gram-negatives). Please comment.

We believe that this apparent ambiguity for Gram negatives is due to efflux pumps. We added MICs performed on *E. coli* strains with knocked out efflux pumps (N43) and membrane permeable *E. coli* (D22) in Table 1. These clearly show that our compounds are effectively pumped by efflux pumps, while membrane permeability is not an issue.

8) Line 187

Check enzyme conc for assays: IC₅₀ should not be below 1/2 of enzyme conc.

All enzyme concentrations are adequate. No IC₅₀ is below ½ of enzyme concentration.

S. aureus = ~ 11.6 nM

S. aureus = ~3 nM (for compounds **5** and **6** determined by gel-based assays)

E. coli = ~3.5 nM

Human topo II = ~1.5 nM

9) *Lines 195-196*

7 and 8 can't perform 2 H-bonds simultaneously in the required directionalities. Also there seems to be not much space for the carbonyl of the amide in 8? Were other NH₂ or S-containing moieties, small aromatic heterocycles considered for potential interactions with the two Ala 68 carbonyl groups?

Our intention at the beginning was not to make two H-bond concomitantly. This is a path that hasn't been explored yet and we plan to investigate this further. The docking data (not included) indicated formation of H-bonds.

19) *Line 197*

How can this be explained? Sterically there seems to be little room for 2 methyl groups?

According to our docking calculations there is enough space for 2 methyl groups. Apparently, each of Ala68 is forming an unusual hydrogen bond with one methyl group, which could be the reason for high activity. Similar interaction was described by Bax et al. (Type IIA topoisomerase inhibition by a new class of antibacterial agents. Nature 466, 935–940 (2010)). Binding mode of compound **9** predicted by docking calculations is shown in Figure S7.

20) *Line 233*

Please provide a reference supporting this statement.

We have cited two papers to support the statement:

1) Chan P. F., et al. Thiophene antibacterials that allosterically stabilize DNA-cleavage complexes with DNA gyrase. Proc. Natl. Acad. Sci. USA. 114, E4492-E4500 (2017)., and

2) Bax, B. D., Murshudov, G., Maxwell, A., Germe, T. DNA Topoisomerase Inhibitors: Trapping a DNA-Cleaving Machine in Motion. J Mol Biol. 431, 3427-3449 (2019).

The text in the manuscript is now equipped with citations (numbers appear in the text instead): “Using ciprofloxacin-induced cleavage as a reference (Chan et al., 2017), we have found that up to approximately 25 % of complexes are cleaved. This is consistent with other NBTI compounds that bind in the same pocket (Bax et al., 2019).”

21) Line 234 (line 221), Fig. 3a

Insufficient labeling:

Conc of compound used?

Lane with no compound

Lane with no enzyme control

Lane showing sizes of linear (cut) DNA

Lane showing size of nicked DNA

We have re-done figure 3. Additionally, Fig. S10 was included in SI for more information.

22) Line 262, Fig. 4

In this particular orientation the asymmetry is not very obvious to me. Maybe there is a better orientation to make the point here?

We believe that the asymmetry is clearly depicted in the Figure 4 b by the asymmetric electron density in the LHS part of the molecule, while the point of Figure 4a is not to present the asymmetry. This is also presented in the figure legend. Please also see the response to the Reviewer 3's first comment.

23) *How does the Fo-Fc density look in this region? What are the B-factors of the ligand atoms in comparison with the surrounding protein atoms?*

Could it be the case that the observed electron density is actually a result of a static disorder of the lower part of the molecule, similar to what was observed for the previous compounds, with slightly different orientations of the aromatic system and ultimately the Cl so that there is actually only one halogen bond with one GyrA subunit per molecule, but randomly distributed within the crystal?

Please see the response to the Reviewer 3's first comment.

24) Line 273

Concerning "asymmetry": Was it built also in the other orientation into the density? (or how does the superposition look?) The orientation of the density in the picture 4b could profit from a superpositioning to show why it is significantly different in the structure reported here, compared to all previous ones.

Please see the response to the Reviewer 3's first comment.

25) Line 311

How does the quality of the electron density look in this particular region (esp. 5b)?

We believe that the quality of the electron density is already well- presented in Fig. S5, which clearly shows the asymmetry. A figure simultaneously showing the densities associated with the two regions when overlaid, as in Fig. 5b, would be difficult to interpret.

26) Line 333-334

To make this statement, data have to be absolutely clear, see comment made for line 234

We have overhauled figure 3. We believe the data clearly show single-strand cleavage stabilisation, as observed for previous NBTIs. Furthermore, we have omitted the word “undoubtedly” in the manuscript, which might be too pushy.

27) Line 370

Could that low occupancy come from residual static disorder, or is it really a second Mn that is partially present additionally to the first Mn?

This is conceivable. However, there is 5 mM MnCl₂ in the crystallisation, therefore having both sites occupied simultaneously is a possibility as they are not mutually exclusive in this configuration. In addition, the refined B-factor of the B-site Mn is comparable to those of its liganding atoms, so a close to unit occupancy is justified for this atom.

39) Line 390

In this paper: <https://doi.org/10.1016/j.comptc.2014.09.024>

A bifurcated halogen bond with a symmetric partner is discussed and the halogen bond is not symmetrical and divided into a primary and secondary interaction. Could it be the case, that the Cl discussed here is either oscillating between the 2 states, or is statically disordered in the crystal?

Please see the response to the Reviewer 3’s first comment. We agree that there is a possibility of the chlorine static disorder, but our data point to the symmetrical halogen bond. Another important thing to take into account is the comparison of Br/I derivatives. Bromo derivative (**5**) turned out to be even more potent than the iodo derivative (**6**). Should the classical or even asymmetrical bifurcated halogen bond take place, we would probably observe the increase in potency for iodo derivative, but this was not the case. In fact, we believe that the binding site is so sterically constrained that very little space would be available for Cl wobbling, as the binding site is just the right size for bromo and likely too tight for iodo substituent. Please also refer to the QM calculations and the answer to your last issue no. 42.

Furthermore, a “pure symmetrical” halogen bond is practically non-existing as in our CSD query both distances vary slightly. Nevertheless, other papers (DOI: 10.1039/C4CP05532B) consider symmetrical bifurcated halogen bonds despite slight difference in the distance.

40) Line 394-396

Does the quality of the local electron density allow this statement? (see comment above)

We believe it does. Please see the response to the Reviewer 3's first comment for details.

41) Line 397

While intuitively a halogen bond is best at 180°, similar to a hydrogen bond interaction, a survey of the PDB/CSD paints a different picture from experimental data. I find the argument from Scholfield et al. compelling that the increased surface of the halogen at 160-165° leads to the 'best' balance between strongest positive point (180°) and strongest negative point (90°) on the ESP of the halogen.

I would highly encourage the authors to come up with a stronger/more definite statement than »should be close to« either by experimental observation (preferred; can of course come from literature) or from QM calculations.

e.g. »To the best of our understanding.. » or »According to experiments...«, »the current knowledge in the field suggest that..«

Again cited from PROTEIN SCIENCE VOL 22:139—152:

»Geometry of X-bonds: The basic concept of the r-hole makes the X-bond a highly directional interaction, as reflected in the angle of approach of the X-bond acceptor to the halogen relative to the direction of the r-bond (H1, Fig. 1). Surveys of H1 angles for small molecule structures in the Cambridge Database²⁹ as well as biomolecular structures in the PDB⁹ indicate a strong preference for a near linear approach of the acceptor toward the electropositive crown of the r-hole, with a significant drop-off as the acceptor approaches the crossing point between the positive and negative electrostatic potentials (H1 ? 140?).«

Especially:

»The balance between the maximum positive electrostatic potential at H1 ¼ 180° with the increase in available surface area of the halogen atom as H1 approaches 90° accounts for the preference for H1 ? 160°–165°.«

Or from Chem. Rev. 2016, 116, 2478–2601 page 2486, 1.4.1 Directionality: »...scatterplots of intermolecular C–X•••N interaction versus X•••N distance (X = I, Br, and Cl). Clearly, short and strong XBs are more directional than the long and weak ones, and by reducing the polarizability of the XB donor, the linearity slightly drops (mean values for the C–X•••N angle are 171.4° for I, 164.1° for Br, and 154.6° for Cl). This trend is general and has also been observed when XB acceptor sites other than nitrogen are used.«

Our survey of reported crystal structures in CSD does not support this Gauss-like distribution and the clear “sweet spot” of Θ_1 and Θ_2 angles. In fact it seems that geometric properties of the molecules involved in halogen bonding rarely allow achieving the “best balance” of 160-165° for Θ_1 . This is why it is hard to make very definite statements since the best balance is not so evident from the statistic analysis/scattergram.

42) Lines 414-416

Since QM calculations are within the scope of the authors (looking at the GUASSIAN09 section in the methods) and the group clearly has a nice crystal structure why not run a QM calculation to get the individual parts of interaction energies from the two carbonyls and the halogen?

I would suggest to run a QM calculation based on the obtained crystal structure with 1.) both carbonyls, 2.) carbonyl only from Ala68 B, 3.) only from Ala68 D, (maybe even one calculation without halogen) to gauge how much each carbonyl contributes to the sum of interactions.

This would in my mind add to the manuscript another (strong) argument.

Advice on QM is clearly not needed but authors might find this also interesting: <https://arxiv.org/pdf/1708.09244.pdf>, Pages 19-25

Also calculations of modeled, asymmetric, more classical bifurcated halogen bonding would be interesting for comparison.

The QM calculation of a modelled system based on the crystal structure was done (see answer 6 to Reviewer #1), and shows a rather minimal difference in the interaction energy of individual carbonyls. Our calculations further demonstrate that the interaction energies are not additive, which gives strong evidence about the role of electron polarization effects. The fact that interaction energies for both Ala68 residues are basically the same despite that carbonyl oxygen atom and chlorine atom distances are slightly different (3.85 Å and 3.61 Å, respectively) gives some evidence that both experimental geometrical parameters of the binding site are either not perfectly accurate and that there is still space for improvement, or that there is indeed a slight distance difference in two halogen bonds, since the interaction energy does not vary markedly by altering the distance even outside of the classical halogen bond distance. This is somehow expected, as halogen are described as either electrostatic interactions or covalent charge-transfer interactions or both (<https://www.nature.com/articles/s41467-020-17122-7>). By increasing the distance, electrostatic part takes place, which is less sensitive to the distance.

The calculations of modelled asymmetric classical bifurcated halogen bonding was also done starting from a MD snapshot (Figure below), and the results are next:

	Eint [ALA68B]	Eint [ALA68D]	Eint [total]

	(kcal mol ⁻¹)	(kcal mol ⁻¹)	(kcal mol ⁻¹)
4 (X-ray)	-2.00	-2.02	-4.23
4 (from MD snapshot)	-1.74	-2.95	-4.88

The calculations of the halogen bond alone show that asymmetric classical bifurcated halogen is expected to be even more favourable, but at the cost of the whole molecule distortion in the aromatic part of the RHS. Namely, the QM calculations we made focus on isolated halogen bond (see Fig. S8) only and do not take into account other parts of the molecule. A mere visual inspection of the starting MD snapshot show an unfavourable distortion of the -NH-C₆H₄-Cl RHS (see Figure b) since it is not planar as expected for an aromatic. Accordingly, the calculation of interaction energies of only halogen bonding part would be misleading and we propose not to include it in the manuscript.

Figure: Comparison of halogen bond parameters of crystallized compound 4 and compound 4 predicted by MD simulation with Ala68. a) Halogen bonding in crystal complex of **4** with *S. aureus* DNA gyrase and DNA. **b)** Halogen bonding of compound **4** predicted by MD simulation (a snapshot with asymmetrical bifurcated halogen bond is presented) within the *S. aureus* DNA gyrase NBTI-binding site of our crystal structure. **c)** Overlap of both structures. Halogen bonds are represented as yellow dots, with interatomic X...O distances [Å] and Θ_1 [C-X...O], i.e., Θ_2 [X...O=C] angles.

Reviewer #1 (Remarks to the Author):

The authors present a fully revised version of their manuscript with an overall substantial improvement compared to their initial submission. All my questions, suggestions, and remarks were addressed satisfyingly and additional references were added.

In detail, the authors have added comments on the non-existing halogen bonding properties of fluorine residues. This is now well explained and informs the reader about the trend of halogen bonding with the higher halogens. Moreover, two detailed databank searches were added. 1) A small molecule search on bifurcated XB motifs accompanied with geometrical and statistical analyses of the search results. Here, the reader is informed about the stringent angle criteria of XB at short contact distances (Fig S13). 2) A PDB search revealed several bifurcated halogen bonding motifs with the help of results cited as Ref 8 of the SI (by Shinada et al). Especially the new Figure S11 provides an informative outline for the reader where such bifurcated XB interactions have been observed (but probably neglected) in earlier work. This figure reveals also how rare the true bifurcated XB motifs are. While Figure panels S11 (f) and (g) show not a fully convincing bifurcated XB, but rather a binding mode of the carboxylates' pi-system to the sigma-hole of the halogen residue (common XB interaction), the other panels impressively demonstrate the structural manifold of this motif. The motif in panel (a) might be an interesting amino acid residue to target with halogens in future studies: the atom sequence of a cysteine with its adjacent amid carbonyl oxygen (...O=C-NH-C(H)-CH₂-S...) could be an interesting topic of further investigations in that context. Similarly interesting is the combination of pi-system XB acceptors and lone-pair XB acceptors in such bifurcated halogen bonds (panels of Figure S11 c–e). The diversity of this often overseen interaction motif, as demonstrated by the newly added data mining, further demonstrated the impact of the presented manuscript.

I recommend publication of the current manuscript in Nature Communications. This will be an impactful contribution for a broad readership of supramolecular chemistry, medicinal chemistry, computational chemistry (such as docking software developers), and in general to the research labs in the pharmaceutical industry.

Reviewer #2 (Remarks to the Author):

All my comments and suggestions were properly addressed in the revised manuscript. I found the authors' responses satisfactory and the manuscript has been much improved. I'd be happy to recommend the paper for publication in Nature Communications.

Reviewer #3 (Remarks to the Author):

Dear Author(s),

Thanks a lot for your careful revision and detailed feedback on the points raised in the first round. There are no more comments on top of the ones already made.

Congratulations to this interesting article.

Author's Response

We are sincerely grateful for all the reviewer's expert advices and remarks, which made the subsequent improvements to our manuscript possible.

Author's Response to Reviewer #1:

The authors present a fully revised version of their manuscript with an overall substantial improvement compared to their initial submission. All my questions, suggestions, and remarks were addressed satisfyingly and additional references were added.

In detail, the authors have added comments on the non-existing halogen bonding properties of fluorine residues. This is now well explained and informs the reader about the trend of halogen bonding with the higher halogens. Moreover, two detailed databank searches were added. 1) A small molecule search on bifurcated XB motifs accompanied with geometrical and statistical analyses of the search results. Here, the reader is informed about the stringent angle criteria of XB at short contact distances (Fig S13). 2) A PDB search revealed several bifurcated halogen bonding motifs with the help of results cited as Ref 8 of the SI (by Shinada et al). Especially the new Figure S11 provides an informative outline for the reader where such bifurcated XB interactions have been observed (but probably neglected) in earlier work. This figure reveals also how rare the true bifurcated XB motifs are. While Figure panels S11 (f) and (g) show not a fully convincing bifurcated XB, but rather a binding mode of the carboxylates' pi-system to the sigma-hole of the halogen residue (common XB interaction), the other panels impressively demonstrate the structural manifold of this motif. The motif in panel (a) might be an interesting amino acid residue to target with halogens in future studies: the atom sequence of a cysteine with its adjacent amid carbonyl oxygen (...O=C-NH-C(H)-CH₂-S...) could be an interesting topic of further investigations in that context. Similarly interesting is the combination of pi-system XB acceptors and lone-pair XB acceptors in such bifurcated halogen bonds (panels of Figure S11 c–e). The diversity of this often overseen interaction motif, as demonstrated by the newly added data mining, further demonstrated the impact of the presented manuscript.

I recommend publication of the current manuscript in Nature Communications. This will be an impactful contribution for a broad readership of supramolecular chemistry, medicinal chemistry, computational chemistry (such as docking software developers), and in general to the research labs in the pharmaceutical industry.

We would like to thank the Reviewer #1 for his/her very positive and courteous feedback on this study. We have modified the previous Supplementary Figure 11 by excluding panels f) and g). Indeed, these were showing π -sigma-hole type of interactions instead of the bifurcated halogen interactions.

Author's Response to Reviewer #2:

All my comments and suggestions were properly addressed in the revised manuscript. I found the authors' responses satisfactory and the manuscript has been much improved. I'd be happy to recommend the paper for publication in Nature Communications.

We would like to thank the Reviewer #2 for his/her very positive and agreeable feedback on this study.

Author's Response to Reviewer #3:

Dear Author(s),

Thanks a lot for your careful revision and detailed feedback on the points raised in the first round. There are no more comments on top of the ones already made.

Congratulations to this interesting article.

We would like to thank the Reviewer #3 for his/her very positive and agreeable feedback on this study.